JCB Journal of Cell Biology

# Critical role of mitochondrial ubiquitination and the OPTN–ATG9A axis in mitophagy

Koji Yamano[1], Reika Kikuchi[1], Waka Kojima[1,2,3], Ryota Hayashida[1,2,3], Fumika Koyano[1], Junko Kawawaki[1], Takuji Shoda[4], Yosuke Demizu[4], Mikihiko Naito[5], Keiji Tanaka[3], and Noriyuki Matsuda[1]

**Damaged mitochondria are selectively eliminated in a process called mitophagy. Parkin and PINK1, proteins mutated in Parkinson's disease, amplify ubiquitin signals on damaged mitochondria with the subsequent activation of autophagic machinery. Autophagy adaptors are thought to link ubiquitinated mitochondria and autophagy through ATG8 protein binding. Here, we establish methods for inducing mitophagy by mitochondria-targeted ubiquitin chains and chemical-induced mitochondrial ubiquitination. Using these tools, we reveal that the ubiquitin signal is sufficient for mitophagy and that PINK1 and Parkin are unnecessary for autophagy activation per se. Furthermore, using phase-separated fluorescent foci, we show that the critical autophagy adaptor OPTN forms a complex with ATG9A vesicles. Disruption of OPTN–ATG9A interactions does not induce mitophagy. Therefore, in addition to binding ATG8 proteins, the critical autophagy adaptors also bind the autophagy core units that contribute to the formation of multivalent interactions in the de novo synthesis of autophagosomal membranes near ubiquitinated mitochondria.**

## Introduction

Parkin and PINK1, the two proteins mutated in familial Parkinson's disease, play critical roles in the selective ubiquitination of damaged mitochondria, which leads to autophagic elimination in a process called mitophagy (Herhaus and Dikic, 2015; Nguyen et al., 2016; Pickrell and Youle, 2015; Yamano et al., 2016). In response to a loss in the mitochondrial membrane potential, PINK1, which is rapidly degraded by the proteasome under normal conditions (Deas et al., 2011; Greene et al., 2012; Jin et al., 2010; Yamano and Youle, 2013), accumulates on the outer mitochondrial membrane (OMM; Hasson et al., 2013; Lazarou et al., 2012; Matsuda et al., 2010; Narendra et al., 2010b) and phosphorylates ubiquitin (Ub) at S65 (Kane et al., 2014; Kazlauskaite et al., 2014; Koyano et al., 2014; Ordureau et al., 2014). Direct interactions between the phosphorylated Ub and the E3 Ub ligase Parkin (Kazlauskaite et al., 2015; Kumar et al., 2015; Sauvé et al., 2015; Wauer et al., 2015; Yamano et al., 2015) recruit cytosolic Parkin to damaged mitochondria (Narendra et al., 2008), where PINK1 phosphorylates S65 in the Ub-like domain of Parkin to activate Parkin E3 ligase activity (Kondapalli et al., 2012; Shiba-Fukushima et al., 2012). The subsequent conjugation of poly-Ub chains to a number of OMM proteins by Parkin (Chan et al., 2011; Sarraf et al., 2013) acts as a

seed for the next round of PINK1-mediated phosphorylation of Ub. This positive feedback amplification loop can thus rapidly coat damaged mitochondria with Ub chains (Okatsu et al., 2015; Ordureau et al., 2014; Shiba-Fukushima et al., 2014). In this model, although PINK1 acts upstream of Parkin, the accumulation of PINK1 can also activate mitophagy independently of Parkin, albeit to a lesser extent (Lazarou et al., 2015). However, whether PINK1-phosphorylated Ub itself is vitally necessary for activating the autophagic machinery remains unclear (Heo et al., 2015; Lazarou et al., 2015; Ordureau et al., 2015; Richter et al., 2016). These controversial issues may arise from the experimental limitation that the Parkin–PINK1 system is currently the only means of inducing the robust ubiquitination of mitochondria. Consequently, to fully answer these questions, a novel method for coating mitochondria with Ub needs to be established.

Macroautophagy (referred to hereafter as autophagy) is a conserved eukaryotic system for mediating the degradation of intracellular components (Lahiri et al., 2019; Mizushima et al., 2011; Morishita and Mizushima, 2019). Many autophagy proteins that form diverse functional units have been identified in mammals. The ULK complex, comprised of FIP200, ATG13, ULK1/2,

[1]Ubiquitin Project, Tokyo Metropolitan Institute of Medical Science, Tokyo, Japan; [2]Department of Computational Biology and Medical Sciences, Graduate School of Frontier Sciences, The University of Tokyo, Chiba, Japan; [3]Laboratory of Protein Metabolism, Tokyo Metropolitan Institute of Medical Science, Tokyo, Japan; [4]Division of Organic Chemistry, National Institute of Health Sciences, Kanagawa, Japan; [5]Division of Molecular Target and Gene Therapy Products, National Institute of Health Sciences, Kanagawa, Japan.

Correspondence to Koji Yamano: yamano-kj@igakuken.or.jp; Noriyuki Matsuda: matsuda-nr@igakuken.or.jp.

and ATG101, functions early on in the activation of a phosphatidylinositol 3-kinase complex that consists of BECN1, ATG14, VPS15, and VPS34. The resultant phosphatidylinositol 3-phosphate facilitates the recruitment of effector proteins such as WIPIs to the autophagosomal formation site. Two Ub-like units, the ATG5–ATG12/ATG16L1 complex and phosphatidylethanolamine-ATG8s, are important for elongation of the autophagic membrane as well as the efficient degradation of the inner autophagosomal membrane in lysosomes (Tsuboyama et al., 2016). In addition, ATG9A, a multi-spanning membrane protein, is integrated in small vesicles and is transiently recruited to the autophagosomal formation site (Orsi et al., 2012; Yamamoto et al., 2012). Hierarchical analysis of autophagy proteins during Parkin-mediated mitophagy showed that the ULK complex and ATG9A vesicles can associate with damaged mitochondria to initiate mitophagy independently of each other and independently of ATG8s (Itakura et al., 2012). The molecular mechanisms underlying this process, however, remain poorly elucidated.

In Ub-mediated selective autophagy, which includes Parkin-mediated mitophagy, autophagy adaptors play an essential role in cargo recognition and subsequent autophagic encapsulation (Randow and Youle, 2014). Currently, five autophagy adaptors, termed OPTN, NDP52, p62, NBR1, and TAX1BP1, have been proposed to link ubiquitinated cargo to autophagosomal membranes, since they contain both Ub-binding domains and an ATG8-interacting motif also known as an LC3-interacting region (LIR; Birgisdottir et al., 2013). Mammals have six different ATG8 homologues (LC3A, LC3B, LC3C, GABARAP, GABARAPL1, and GABARAPL2; Johansen and Lamark, 2019). The binding affinities of ATG8s for autophagy adaptors are thought to play a crucial role in recruitment of the autophagic membrane to the cargo. Although all five autophagy adaptors are recruited to the damaged mitochondria during mitophagy, only OPTN and NDP52 are critical for mitochondrial clearance (Heo et al., 2015; Lazarou et al., 2015). The function of OPTN and NDP52 in Parkin-mediated mitophagy was reported to involve recruitment of the upstream autophagy machinery, including the ULK complex (Lazarou et al., 2015). The interaction between NDP52 and FIP200 was subsequently shown to be important in the recruitment of the ULK complex to damaged mitochondria and invading bacteria (Ravenhill et al., 2019; Vargas et al., 2019). The molecular basis for why OPTN is primarily essential for Parkin-mediated mitophagy, however, remains unclear.

## Results

### Mitophagy induced by ectopic mitochondria-targeted Ub chains

It remains a matter of debate whether PINK1 itself or PINK1-generated phosphorylated Ub activates autophagy machinery. To address this issue, we coated mitochondria with poly-Ub chains independent of Parkin or PINK1. For this purpose, different lengths of linear Ub chains were fused to the N-terminal transmembrane segment (1–49 aa) of TOMM20 and YFP (Fig. 1 A). Linear Ub chains can be modified with branched chains by cytosolic E3 ligases (Ohtake et al., 2016). To prevent this, we generated two additional sets of Ub-chain constructs, one with a

K48R mutation in which Ub lysine-48 was replaced with arginine and a second set of constructs (K0 mutations) with all seven lysine residues in Ub replaced with arginine (Fig. 1 A). For Parkin-independent mitophagy, OMM-Ub proteins were transiently expressed in HeLa cells, which do not express endogenous Parkin (Denison et al., 2003). All of the OMM-Ub proteins were precisely targeted to the mitochondria (Fig. S1 A). Additional ubiquitinated forms were also found with the WT and K48R chains (indicated by blue ladders in Fig. 1 B), but not in the K0 chains. When we detected TOMM20 and the matrix protein MTCO2 under these conditions, the levels of both proteins seemed to be reduced in cells expressing OMM-2Ub K0 (Fig. 1 B). When plasmids for OMM-Ub K0 were transfected using Lipofectamine LTX to achieve higher protein expression, a clear reduction in TOMM20 and MTCO2, but not actin, was observed (Fig. 1 C), suggesting mitochondrial degradation.

Next, we used FACS in conjugation with mitochondria-targeted Keima (mt-Keima; Katayama et al., 2011) to detect mitophagy. Mitophagy-induced delivery of mt-Keima to lysosomes induces a spectral shift in Keima due to the lower pH of the lysosome. OMM-Ub proteins were transiently expressed in HeLa cells stably expressing mt-Keima. Similar to the cytosolic YFP alone, the WT and K48R linear Ub-chains did not induce mitophagy (Fig. 1, D and E). In sharp contrast, when the K0 version of OMM-2Ub was expressed, mitophagy occurred in >80% of the cells (Fig. 1, D and E). While less efficient than 2Ub, both 4Ub and 6Ub K0 could also induce mitophagy (Fig. 1, D and E). These results suggest that inhibition of branched chain formation is an important trigger for mitophagy. Next, we used ATG5 knockout (KO) HeLa cells (Nezich et al., 2015) and Penta KO HeLa cells that lack all five autophagy adaptors (Lazarou et al., 2015). The Keima-FACS assay clearly showed an absence of mitophagy in both the ATG5 KO and Penta KO cells, even when the OMM-2Ub K0 or OMM-6Ub K0 proteins were expressed (Figs. 1 F and S1 B). When each autophagy adaptor was reintroduced into Penta KO cells, OPTN, p62 and NBR1, but not NDP52, were recruited to mitochondria in an OMM-2Ub K0–dependent manner (Fig. 1 G). Furthermore, the FACS assay showed that only OPTN expression in the Penta KO cells recovered OMM-Ub K0–induced mitophagy (Figs. 1 H and S1 C). Since OMM-Ub K0 induces mitophagy without any depolarization-induced chemicals that cause PINK1 accumulation, it is assumed that OMM-Ub–induced mitophagy is PINK1 independent. To examine this, we used a PINK1 KO HeLa cell line. Although antimycin A/oligomycin (AO) treatment slightly increased the overall mitophagy rate, mitophagy was not suppressed in the PINK1 KO cell line (Figs. 1 I and S1 D).

### SNIPER-induced mitophagy

In addition to ectopically expressed linear Ub chains, we used synthetic hybrid molecules termed specific and nongenetic inhibitor of apoptosis (IAP)–dependent protein erasers (SNIPERs; Itoh et al., 2010) to induce endogenous E3 ligases to coat mitochondria with Ub chains. One of the SNIPERs we developed, SNIPER(CRABP)-11, is a small hybrid compound linking the RING-type E3 ligase cIAP1 (cellular IAP protein 1) and CRABP-II (cellular retinoic acid binding protein II) such that cIAP1

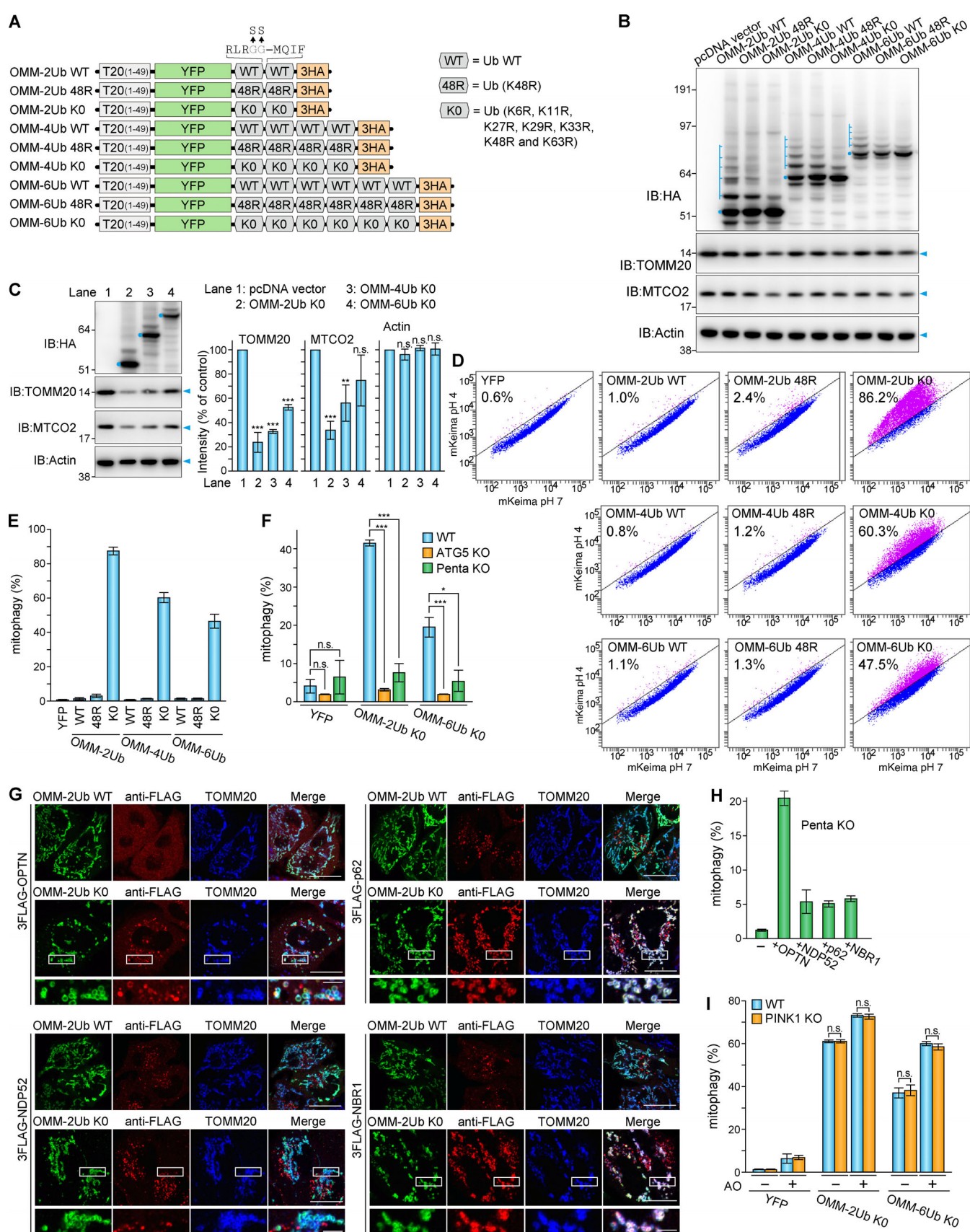

Figure 1. **Mitochondria-targeted linear Ub chains induce mitophagy. (A)** Schematic diagram of mitochondrial targeted linear Ub chains. T20(1–49) denotes the N-terminal 49 aa of TOMM20. G75S/G76S mutations were introduced in the C-terminus of each Ub to block cleavage. **(B)** Immunoblots (IB) of HeLa

cells transiently expressing the indicated proteins. The full-length proteins and those with additional Ub chains are indicated by blue dots and ladders, respectively. **(C)** The indicated proteins were expressed using Lipofectamine LTX. Total cell lysates 30 h after transfection were immunoblotted. Quantification graphs of TOMM20, MTCO2, and actin levels are indicated on the right. Error bars represent mean ± SD of three independent experiments. **(D)** The indicated proteins were transiently expressed in HeLa cells. At 48 h after transfection, cells were analyzed by FACS. Representative FACS data with the percentage of cells exhibiting lysosomal positive mt-Keima are shown. **(E)** Quantification of the FACS-based mitophagy in D. Error bars represent mean ± SD of three independent experiments. **(F)** Quantification of FACS-based mitophagy in WT, ATG5 KO, or Penta KO HeLa cells expressing the indicated proteins. Error bars represent mean ± SD of three independent experiments. **(G)** OMM-2Ub WT or K0 were transiently expressed in Penta KO HeLa cells with stable expression of each 3FLAG-autophagy adaptor. At 24 h after transfection, cells were immunostained. Scale bars, 10 μm; insets, 2 μm. **(H)** Quantification of FACS-based mitophagy using cells in G. Error bars represent mean ± SD of three independent experiments. **(I)** Quantification of FACS-based mitophagy in WT and PINK1 KO HeLa cells expressing the indicated proteins with or without 6 h of AO treatment. Error bars represent mean ± SD of three independent experiments. n.s., not significant; *, P < 0.05; **, P < 0.01; ***, P < 0.001.

mediates ubiquitination of CRABP-II in the presence of SNIPER(CRABP)-11 (Itoh et al., 2012; Fig. 2 A). To induce the association of Ub chains on mitochondria, we screened a number of OMM proteins as candidates for fusion with CRABP-II before selecting hexokinase-1 (HK1) and TOMM20 (Fig. 2 B). The fused proteins and 3HA-tagged CRABP-II alone were expressed in human fibrosarcoma HT1080 cells, which have elevated expression levels of endogenous IAP family E3 ligases (Fig. 2 C). Both CRABP-II–fused HK1 (HK1-CII-3HA) and TOMM (T20-CII-3HA) localized on mitochondria (Fig. 2 D). SNIPER-induced ubiquitination promoted degradation of the CRABP-II fusion proteins (Fig. 2 E). Under these conditions, lysosomal mt-Keima signals were observed in cells expressing HK1-CII-3HA in a SNIPER-dependent manner (Fig. 2 F). The FACS-based assay also showed that mt-Keima spectral shifts were clearly observed in cells expressing HK1-CRABP-II-3HA (Fig. 2, G and H). While T20-CII-3HA–expressed cells had slightly higher mitophagy rates in the absence of SNIPER(CRABP)-11, the SNIPER-dependent Keima shift was also observed. Importantly, these Keima shifts were neutralized by bafilomycin A1, an inhibitor of lysosome function (Fig. 2, G and H). Next, we knocked down PINK1 in HT1080 cells by siRNA to determine if SNIPER-induced mitophagy depends on PINK1. We first confirmed PINK1 knockdown by both immunoblotting and a deficiency in the translocation of GFP-Parkin (Figs. 2 I and S2). Under these PINK1-knockdown conditions, the efficiency of SNIPER-induced mitophagy in cells expressing HK1-CII-3HA was similar to that in cells treated with the control siRNA (Fig. 2, J and K).

These results indicate that mitochondrial ubiquitination by either ectopic linear Ub chains or the SNIPER method is sufficient for inducing mitophagy. Under these conditions, both Parkin and PINK1 are dispensable for mitophagy, indicating that the Parkin–PINK1 system is essential for ubiquitination of damaged mitochondria but is not required for autophagy activation per se.

### Heterogeneous mitochondrial localization of NDP52 and OPTN

Two autophagy adaptors, NDP52 and OPTN, are critical for Parkin-mediated mitophagy (Heo et al., 2015; Lazarou et al., 2015). The mechanism underlying their function, however, remains to be elucidated. To gain insights into this process, we carefully observed the mitochondrial localization of autophagy adaptors during Parkin-mediated mitophagy. HeLa cells stably coexpressing each of the autophagy adaptors with a 3FLAG tag and Parkin were treated with valinomycin for 3 h. Since

TAX1BP1 did not express well in our HeLa cells, we focused on the other four autophagy adaptors. All of the 3FLAG-tagged autophagy adaptors (Fig. 3 A) were recruited to the mitochondria in response to Parkin-mediated ubiquitination (Fig. 3 B). Interestingly, OPTN and NDP52 were heterogeneously recruited to mitochondria, while p62 and NBR1 were evenly distributed (Fig. 3 B). Furthermore, endogenous p62 and 3FLAG-NBR1 were colocalized during mitophagy, whereas OPTN and NDP52 only partially overlapped with endogenous p62 (Fig. 3 C). The critical mitophagy adaptors (OPTN in particular) preferentially bound the K63-linked Ub chains (as well as the linear Ub chains). Therefore, we next observed the mitochondrial localization of different Ub linkages (K48- and K63-linked chains). Ub signals detected using anti-FK2 (for multi/mono-Ub), anti-Apu2 (K48-linked chains), or anti-Apu3 (K63-linked chains) antibodies were homogeneously distributed on the mitochondria (Fig. 3 D). Consequently, mitochondrial accumulation of linkage-specific Ub chains cannot account for the heterogeneous recruitment of OPTN and NDP52. Since autophagy adaptors contain LIR, OPTN and NDP52 may be recruited to contact sites between mitochondria and the autophagosomal formation site. Indeed, signals for OPTN and NDP52 were concentrated on mitochondria in close proximity to LC3B (Fig. 3 E), suggesting that OPTN and NDP52, but not p62 and NBR1, are preferentially assembled within a particular region of the mitochondria where formation of the LC3-positive autophagosomal membrane occurs.

### Generation of phase-separated foci composed of linear Ub chains and autophagy adaptors

We hypothesized that LIR motifs in OPTN and NDP52 may have higher binding affinities for ATG8 family proteins than those in p62 or NBR1. Indeed, a recent report indicates that interactions between ATG8 proteins and the LIR motif in OPTN (and NDP52) drive additional recruitment of OPTN (and NDP52) to the growing autophagic membrane through an ATG8-dependent positive feedback loop (Padman et al., 2019). To monitor the binding of ATG8 proteins to the autophagy adaptors in cells, we used the protein–protein interaction technology termed Fluoppi (Koyano et al., 2014; Watanabe et al., 2017; Yamano et al., 2015). An Ash tag, which forms a homooligomer, was fused to linear 6Ub (WT or K0), and a homotetrameric humanized Azami-Green (hAG) tag was fused to the autophagy adaptors (Fig. 4 A). Through multivalent interactions between Ub and Ub-binding domains, hAG forms phase-separated fluorescent foci (referred to hereafter as Fluoppi foci) in cells. If there is a direct

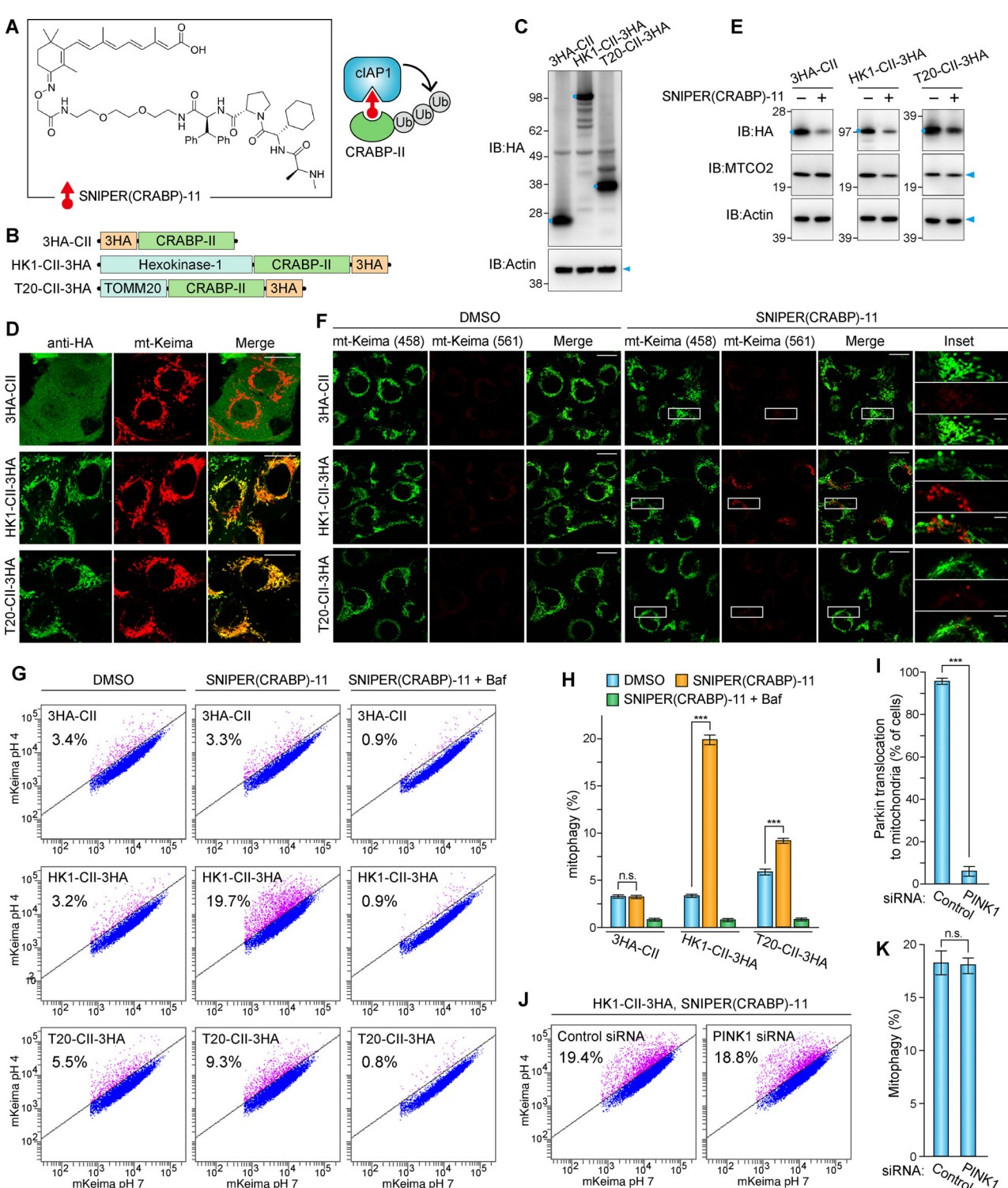

Figure 2. **SNIPER-induced mitophagy independent of PINK1 function. (A)** Chemical structure of SNIPER(CRABP)-11 and schematic diagram of CRABP-II ubiquitination. **(B)** Schematic diagram of the CRABP-II (CII) fusion with HK1 and TOMM20 (T20). **(C)** Immunoblots of HT1080 cells stably expressing the indicated proteins. The full-length CRABP-II–fused proteins are indicated by the blue dots. **(D)** HT1080 cells stably expressing mt-Keima and the indicated proteins were immunostained. Scale bars, 10 μm. **(E)** Immunoblots of HT1080 cells stably expressing the indicated CRABP-II proteins following treatment with or without SNIPER(CRABP)-11 for 9 h. The full-length CRABP-II-fused proteins are indicated by the blue dots. **(F)** Live-cell confocal microscopy imaging of HT1080 cells in D treated with or without SNIPER(CRABP)-11 for 9 h. mt-Keima (561) signals denote mitochondria in lysosomes. Scale bars, 10 μm; insets, 2 μm. **(G)** HT1080 cells stably expressing mt-Keima and the indicated proteins were treated with or without SNIPER(CRABP)-11 for 9 h. Bafilomycin A1 (Baf) was added 3 h before FACS analysis. Representative FACS data with the percentage of cells exhibiting lysosomal positive mt-Keima are shown. **(H)** Quantification of FACS-based mitophagy in G. Error bars represent mean ± SD of three independent experiments. **(I)** HT1080 cells stably expressing GFP-Parkin were treated with control or PINK1 siRNA. GFP-Parkin translocation to the mitochondria was induced by 3hr treatment with valinomycin. The percentage of

cells with Parkin translocation was quantified. Error bars represent mean ± SD, with >100 cells counted in each of three independent experiments. **(J)** HT1080 cells in G treated with control or PINK1 siRNA were treated with SNIPER(CRABP)-11 for 9 h and then analyzed by FACS. Representative FACS data with the percentage of cells exhibiting lysosomal positive mt-Keima are shown. **(K)** Quantification of the FACS-based mitophagy in J. Error bars represent mean ± SD of three independent experiments. n.s., not significant; ***, P < 0.001.

autophagy adaptor interactant, it would also be incorporated into the Fluoppi foci, and autophagosomal membrane formation would be expected to be observed near the foci (Fig. 4 A). Consistent with our expectation, we were able to generate Fluoppi foci consisting of hAG-tagged autophagy adaptors and Ash-tagged 6Ub (Fig. 4 B). Immunoblotting showed that the protein levels of HA-Ash-6Ub increased when expressed with the hAG-tagged autophagy adaptors (Fig. 4 C), suggesting that the autophagy adaptors shift HA-Ash-6Ub from proteasomal degradation to autophagic cargo formation. In addition, OPTN overexpression increased levels of the activated form of TBK1 (detected as S172 phosphorylation by immunoblotting), which was further enhanced by coexpression with HA-Ash-6Ub (Fig. 4 C). This is consistent with previous results demonstrating that OPTN and Ub chains facilitate positive feedback activation of TBK1 (Heo et al., 2015).

Using the Fluoppi assay, we sought to determine if OPTN and NDP52 bind ATG8 family proteins better than p62 and NBR1. An ATG5 KO cell line was used, because we wanted to focus on direct interactions between the autophagy adaptors and ATG8 proteins in cells without the potential involvement of any autophagy machinery. Fluoppi foci were produced in ATG5 KO cells with the stable expression of 3FLAG-tagged ATG8 proteins (Fig. 4, D and E), and the efficiency of their recruitment to the foci was quantified (Fig. 4, F and G; and Fig. S3). As expected, lipidation of ATG8 proteins was completely blocked in ATG5 KO cells (Fig. 4, D and E). Unexpectedly, the incorporation of ATG8 proteins into the OPTN and NDP52 Fluoppi foci was lower than that of the p62 or NBR1 Fluoppi foci (Fig. 4, F and G). Furthermore, quantification values for an OPTN S177D phosphomimetic mutant that was previously shown to tightly bind LC3B (Wild et al., 2011) did not reach those of p62 or NBR1 (Fig. 4 G). These results strongly suggest that the critical function of OPTN and NDP52 in mitophagy is not primarily derived from their binding affinity for ATG8 proteins.

### OPTN Fluoppi foci contain ATG9A vesicles
Recently, NDP52 interactions with FIP200 have been reported to affect ULK complex recruitment to damaged mitochondria and invading bacteria (Ravenhill et al., 2019; Vargas et al., 2019). Similar to NDP52, OPTN may bind autophagy core subunits in addition to ATG8 proteins. To examine this possibility, we continuously used the Fluoppi assay. TBK1, previously shown to directly interact with OPTN, was incorporated into the OPTN foci (Fig. 5 A). In contrast, ATG13, ATG14, WIPI2, and ATG16L1 localized in small dot-like structures adjacent to the OPTN Fluoppi foci (Fig. 5 A), suggesting that these autophagy proteins do not directly interact with OPTN but are recruited in close proximity to facilitate recognition of the foci as an autophagic cargo. Indeed, LC3B appeared to partially surround the OPTN Fluoppi foci (Fig. 5 A). In sharp contrast, when we

immunostained endogenous ATG9A, the signal completely overlapped with the OPTN foci, strongly suggesting a direct interaction (Fig. 5 A). To rule out the possibility that other endogenous autophagy adaptors contributed to ATG9A incorporation into the OPTN foci, we used Penta KO HeLa cells. The ATG9A content of OPTN Fluoppi foci in the Penta KO HeLa cells was comparable to that in WT HeLa cells (Fig. 5 B) but absent in NDP52, p62, and NBR1 Fluoppi foci (Fig. 5, B and C), indicating a specific interaction between OPTN and ATG9A. To investigate whether other autophagy core subunits affect the interaction between OPTN and the ATG9A vesicles, we repeated the Fluoppi assay with FIP200 KO and ATG5 KO cells (Fig. 5, D and E). In both KO cell lines, the efficiency of ATG9A vesicle incorporation in the foci was similar to that in WT cells (Fig. 5 F). Furthermore, in Penta KO cells, the OPTN Fluoppi foci contained ATG9A, but not ATG13, whereas NDP52 Fluoppi foci contained ATG13, but not ATG9A (Fig. S4 A). We also found that deletion of the ATG9A gene (Fig. 5 D) caused ATG13 puncta formation (Fig. S4 B). Under this condition, the NDP52 foci still completely overlapped with ATG13, whereas the ATG13 puncta were only in close proximity to foci generated by the other adaptors, including OPTN (Fig. S4 B). These results are consistent with previous findings that NDP52 directly recruits the ULK complex in mitophagy (Vargas et al., 2019) and demonstrate the utility of the Fluoppi assay as a novel in-cell method for identifying specific protein–protein interactions in a sequential autophagy cascade.

### The OPTN–ATG9A interaction depends on the leucine zipper domain, but not TBK1 activity or Ub binding
Since OPTN contains several distinct domains (Fig. 6 A), we next used mutational mapping to determine the OPTN domain responsible for interactions with ATG9A. We first introduced an S473E mutation into hAG-OPTN to maintain the higher Ub-binding ability (Richter et al., 2016). N-terminal truncations showed that deletion of 1–149 aa and beyond blocked incorporation of ATG9A vesicles into the OPTN Fluoppi foci (Fig. 6 A and Fig. S5, A and B). The OPTN 1–127 aa deletion mutants lost both physical interaction with TBK1 and TBK1 activation (Figs. 6 A and S5 C). We next sequentially replaced five amino acids across residues 128–152 aa of OPTN with a five-alanine repeat (5A). Although the 5A replacement within the 128–142 aa region did not affect ATG9A vesicle incorporation into the Fluoppi foci, two mutants (143-5A-147 and 148-5A-152) generated Fluoppi foci devoid of ATG9A vesicles (Fig. 6 A and Fig. S5, D and E). All of the 5A mutants, however, still interacted with and activated TBK1 in the foci (Figs. 6 A and S5 F). We noticed that OPTN residues 143–164 aa correspond to a leucine zipper. To disrupt this motif, we substituted the leucines with alanine either individually or in total (4LA). Both sets of substitutions inhibited ATG9A vesicle incorporation into the Fluoppi foci (Fig. 6, A and B; and Fig. S5, G and H) but maintained TBK1 activation (Figs. 6 A

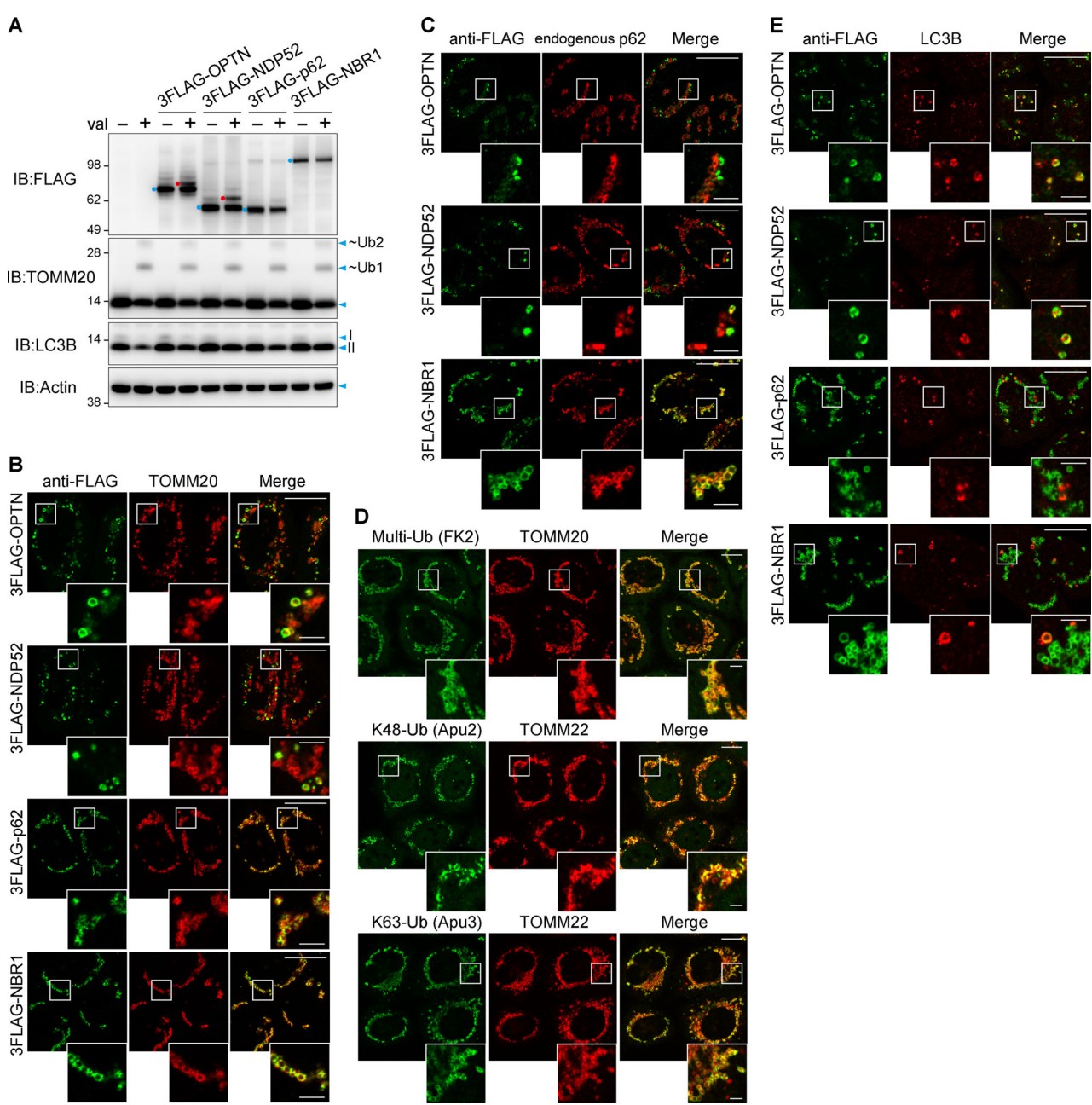

Figure 3. **Heterogeneous mitochondrial localization of OPTN and NDP52 during Parkin-mediated mitophagy. (A)** Immunoblots of HeLa cells stably expressing untagged Parkin with or without 3FLAG-tagged autophagy adaptors (OPTN, NDP52, p62, and NBR1) following valinomycin treatment for 3 h. 3FLAG-tagged autophagy adaptors and those with ubiquitination are indicated by the blue and red dots, respectively. I and II denote unmodified and lipidated LC3B, respectively. ~Ub1 and ~Ub2 denote ubiquitinated forms of TOMM20. **(B)** HeLa cells in A were treated with valinomycin for 3 h and then immunostained. **(C)** The cells prepared in B were immunostained with anti-FLAG and anti-p62 antibodies. **(D)** HeLa cells stably expressing untagged Parkin were treated with valinomycin for 3 h and then immunostained. **(E)** The cells prepared in B were immunostained with anti-FLAG and anti-LC3B antibodies. Magnified images in B–E are shown. Scale bars, 10 µm; insets, 2 µm.

and S5 I). These results indicate that OPTN leucine zipper is required for interactions with ATG9A vesicles.

Next, we evaluated the contribution of TBK1 activity to OPTN-ATG9A binding using either a TBK1 chemical inhibitor BX-795 or OPTN M44Q/L54Q mutants defective for TBK1 interactions (Li et al., 2016). Although neither BX-795 treatment nor the M44Q/L54Q mutation blocked incorporation of ATG9A vesicles into the OPTN Fluoppi foci, the phosphorylated TBK1

signal was completely lost (Fig. 6, C–F). Immunoprecipitation of OPTN Fluoppi foci confirmed microscopic observation. In pull-down assays, ATG9A, but not TBK1, was associated with the M44Q/L54Q hAG-OPTN mutant, whereas the reciprocal association was seen with the 4LA hAG-OPTN mutant (i.e., TBK1 present but not ATG9A; Fig. 6 G). Furthermore, the ATG9A vesicle component of OPTN Fluoppi foci in the TBK1[−/−] HCT116 cells (Fig. 6 H) was comparable to the WT cells (Fig. 6, I and J).

**Figure 4. Incorporation of ATG8 family proteins into phase-separated fluorescent foci composed of autophagy adaptors and linear Ub chains.**
**(A)** Schematic diagram for generating phase-separated fluorescent foci (Fluoppi foci). The direct autophagy adaptor interactant will be incorporated into the Fluoppi foci, and the autophagosomal membrane formation will occur near the Fluoppi foci. **(B)** HeLa cells transiently expressing hAG-tagged autophagy adaptors and HA-Ash-6Ub were immunostained. Scale bars, 20 μm. **(C)** Immunoblots of cells in (B). The full-length of hAG-tagged proteins and HA-Ash–tagged proteins are indicated by blue dots, and ubiquitinated HA-Ash-6Ub are indicated by the red dots. **(D)** Immunoblots of ATG5 KO HeLa cells with or without stable expression of the indicated 3FLAG-ATG8 proteins. Blue dots indicate the unmodified 3FLAG-ATG8 proteins. **(E)** Immunoblots of WT and ATG5 KO HeLa cells stably expressing 3FLAG-LC3B. Blue and red dots indicate the unmodified and lipidated 3FLAG-LC3B, respectively. **(F)** Representative microscopic images of

ATG8 incorporation into Fluoppi foci. Scale bars, 10 µm. **(G)** Efficiency of 3FLAG-ATG8 incorporation into the Fluoppi foci. Error bars represent mean ± SD, with 50 Fluoppi foci quantified in two independent experiments.

These results indicate that OPTN interaction with ATG9A is not affected by TBK1, and vice versa.

Using an FRB-FKBP system in conjugation with an A/C heterodimerizer chemical that induced dimerization of the two components, we were able to localize OPTN to mitochondria (Fig. 6 K). The FRB construct, FRB-FIS-TM, has an FRB domain fused to a transmembrane segment of FIS1 as an OMM anchor, whereas the FKBP portion (OPTN-2FKBP-HA) consists of OPTN with the Ub-binding domains (residues 445–577 aa, which include the UBAN and zinc finger domains) replaced with two tandem FKBPs (Fig. 6 K). Following transient expression in HeLa cells, cytosolic OPTN-2FKBP-HA was translocated to the mitochondria by the addition of the A/C heterodimerizer and subsequently recruited ATG9A vesicles to the mitochondria (Fig. 6, L and M). In contrast, ATG9A vesicles were not recruited to mitochondria when expressed with OPTN-2FKBP-HA harboring the 4LA mutation (Fig. 6, L and M). This further supports the OPTN–ATG9A interaction and demonstrates that the two Ub-binding domains, UBAN and zinc finger, in OPTN are dispensable for interactions with ATG9A vesicles.

### The OPTN–ATG9A interaction is crucial for Parkin-mediated mitophagy

Is the OPTN–ATG9A interaction required for Parkin-mediated mitophagy? To address this question, we expressed 3FLAG-OPTN WT or the 4LA mutant in Penta KO cells. We also expressed OPTN F178A, which has a mutation in the LIR that inhibits ATG8 interactions (Wild et al., 2011), and a double OPTN mutant with 4LA and F178A (4LA/F178A). The Keima-FACS assay showed an absence of mitophagy in the Penta KO cells (Fig. 7, A and B). OPTN WT expression recovered mitophagy (Fig. 7, A and B) as reported previously (Lazarou et al., 2015). In contrast, the 4LA mutant only slightly (18%) and the F178A mutant moderately (53%) recovered mitophagy after 3 h of AO treatment. Of note, mitophagy recovery was almost completely blocked by the 4LA/F178A mutant (Fig. 7, A and B). GFP-Parkin was efficiently translocated to mitochondria in all OPTN variant–expressing Penta KO cells (Fig. 7 C). After 18 h of AO treatment, Penta KO cells rescued with 3FLAG-OPTN WT robustly degraded the matrix proteins MTCO2 and PDH, whereas the OPTN 4LA and 4LA/F178A mutants did not (Fig. 7 D). The OMM proteins MFN2 and TOMM20 were almost completely degraded irrespective of OPTN expression.

Furthermore, although both OPTN WT and the F178A mutant were able to recruit ATG9A to mitochondria in Penta KO cells during Parkin-mediated mitophagy, OPTN harboring the 4LA mutation completely abrogated ATG9A recruitment to the mitochondria (Fig. 7, E and F). Interestingly, while mitochondrial clearance is recovered following reconstitution of NDP52 in Penta KO cells (Lazarou et al., 2015), the robust recruitment of ATG9A to mitochondria was not induced (Fig. 7, E and F). These results suggest that the OPTN–ATG9A interaction is specific in mitochondrial clearance and that this process is facilitated by

interactions with OPTN–ATG8s in concert with ATG9A. Line scan plots indicated that 3FLAG-OPTN WT was heterogeneously recruited to mitochondria in WT HeLa during mitophagy (Fig. 7 G). Although the mitophagy-deficient mutants 4LA and 4LA/F178A still translocated to mitochondria, they were evenly distributed to damaged mitochondria (Fig. 7, G and H), indicating that the mitophagy defect and mitochondrial distribution (i.e., assembly to the autophagosomal formation site) are correlated.

## Discussion

Numerous studies have revealed that Parkin and PINK1 coordinately recognize and ubiquitinate damaged mitochondria. However, whether Parkin and PINK1 function solely in the ubiquitination of the target organelles or there is physical communication with autophagy core machinery remains to be elucidated. In this study, we established a system for coating mitochondria with Ub chains independent of the Parkin and PINK1 pathway. A number of previous studies have sought to induce mitophagy using ectopic mitochondria-targeted Ub. Narendra et al. targeted monomeric Ub to the OMM using an FRB-FKBP system, but no reduction in mitochondrial mass was observed (Narendra et al., 2010a). We previously reported that mitochondrial linear Ub chains function as a Parkin receptor when phosphorylated by PINK1. Although p62 and LC3B were recruited to mitochondria under these conditions (Okatsu et al., 2015), the evidence for mitochondrial elimination was less clear. Zheng et al. showed that linear Ub chains fused to TOMM70 were not sufficient for inducing mitophagy (Zheng and Hunter, 2013). In sharp contrast, the linear chains consisting of Ub K0 that we made in this study prevented branched chain formation and effectively induced mitophagy. Structurally, linear and K63-linked Ub chains adopt equivalent structures, but their ternary structures differ from that of the K48-linked chains (Komander et al., 2009). Indeed, the UBAN domain in OPTN preferentially binds to linear or K63-linked chains. Furthermore, the K63-linked Ub was degraded through proteasomes by K48-linked branched chains (Ohtake et al., 2018). It is clear that linear Ub chains composed of WT Ub are highly modified by branched chains, which shift pathway destiny from autophagy adaptor association to proteasomal degradation. In this study, we also demonstrated that SNIPER-induced mitochondrial ubiquitination by endogenous cIAP E3 ligases can induce mitophagy. Although less effective than Parkin/PINK1-mediated ubiquitination, our data clearly show that SNIPER-induced ubiquitination can be used as a tool to eliminate the target organelles as well.

Once activated, Parkin can ubiquitinate many different OMM proteins, suggesting that Parkin does not possess rigorous substrate specificity (Koyano et al., 2019b; Sarraf et al., 2013). Effects of this polysubstrate ubiquitination include rapid degradation of Mfn1/2, which is thought to be important for segregating damaged mitochondria from the healthy network (Tanaka et al., 2010); increased mitochondrial degradation

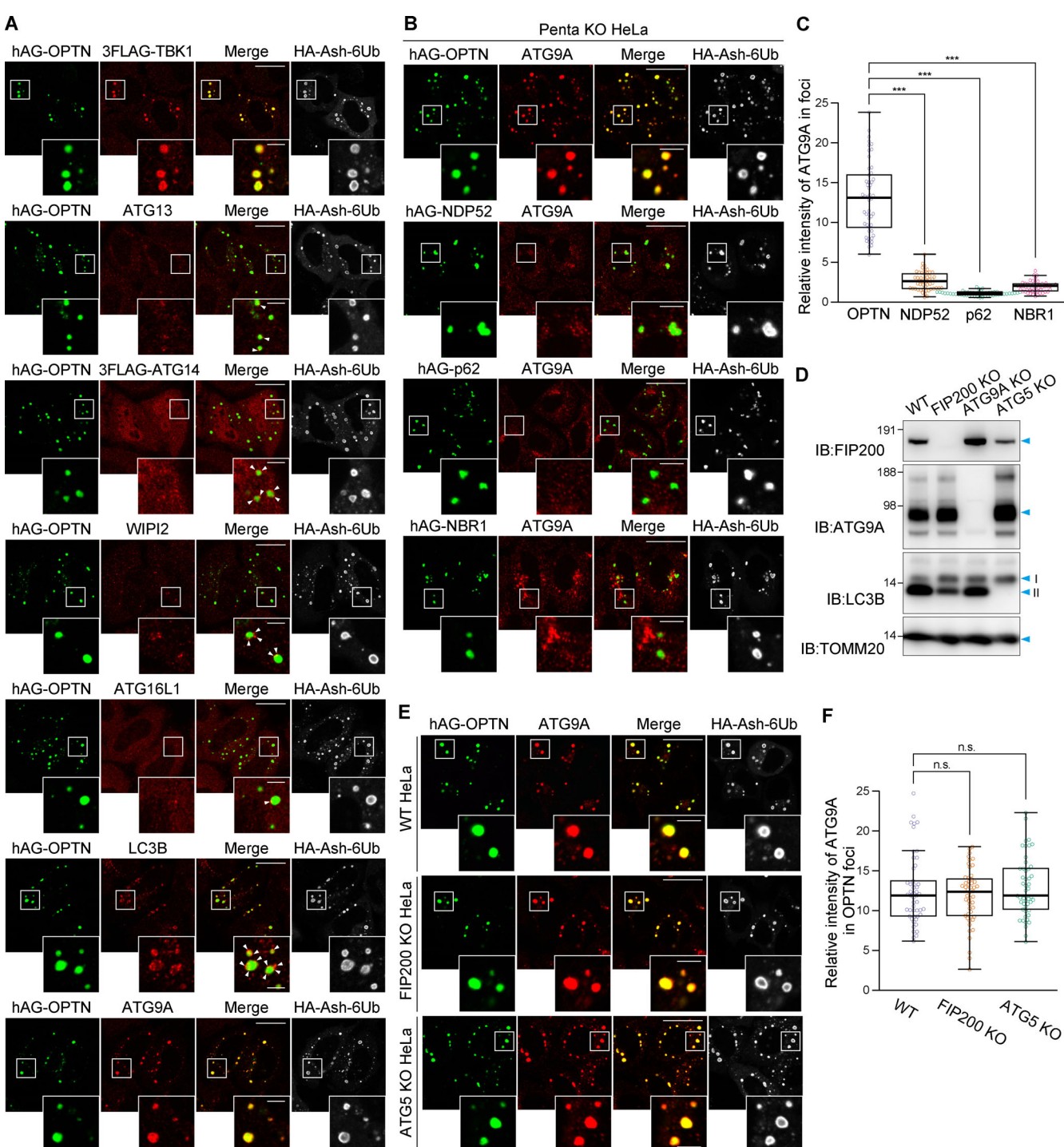

Figure 5. **OPTN Fluoppi foci contain ATG9A vesicles. (A)** HeLa cells expressing hAG-OPTN and HA-Ash-6Ub were immunostained. For the detection of TBK1 and ATG14, HeLa cells stably expressing the 3FLAG-tagged version were used. White arrowheads indicate autophagy proteins recruited in close proximity to the Fluoppi foci. **(B)** Penta KO cells expressing hAG-tagged autophagy adaptors and HA-Ash-6Ub were immunostained. **(C)** Efficiency of ATG9A incorporation into the Fluoppi foci in B. Error bars represent mean ± SD, with 50 Fluoppi foci quantified in two independent experiments. **(D)** Immunoblots confirming KO of FIP200, ATG9A, and ATG5. I and II denote unmodified and lipidated LC3B, respectively. **(E)** WT, FIP200, and ATG5KO KO HeLa cells expressing hAG-OPTN and HA-Ash-6Ub were immunostained. **(F)** Efficiency of ATG9A incorporation into OPTN Fluoppi foci in E. Error bars represent mean ± SD with 50 Fluoppi foci quantified in two independent experiments. Magnified images are shown in A, B, and E. Scale bars, 10 µm; insets, 2 µm. n.s., not significant; ***, P < 0.001.

following termination of mitochondria–ER contacts in response to Mfn2 ubiquitination (McLelland et al., 2018); and rapid Miro1 degradation, which arrests microtubule-dependent mitochondrial trafficking (Wang et al., 2011). While ubiquitination of these OMM proteins facilitate and/or assist efficient mitochondrial degradation, our study revealed that mitochondria-associated Ub chains are sufficient for mitophagy without proteasomal degradation of endogenous OMM proteins.

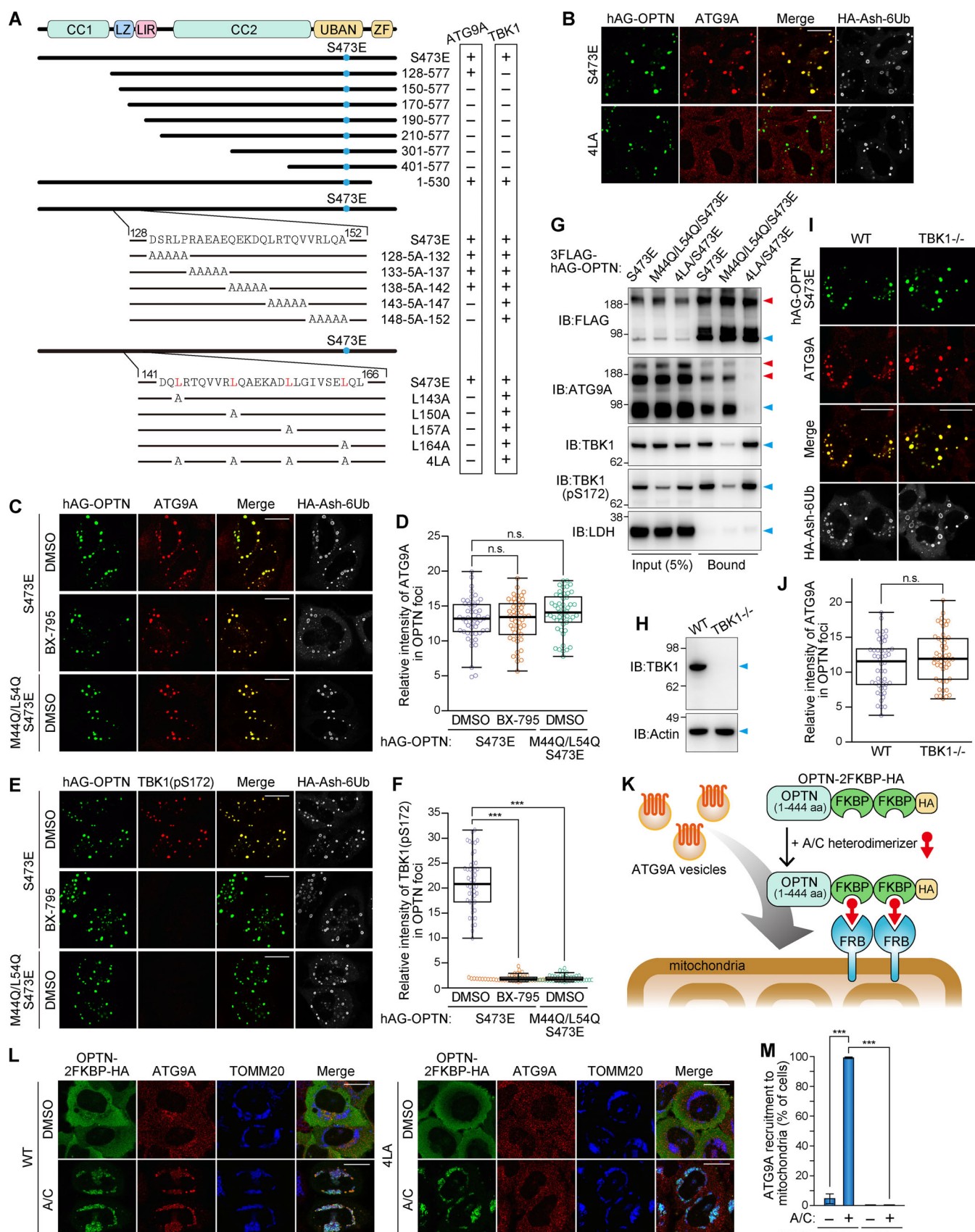

Figure 6. **OPTN–ATG9A interaction depends on the leucine zipper domain, but not TBK1 activity or Ub binding. (A)** Schematic diagram of the OPTN domain architecture and hAG-tagged OPTN mutants. CC, coiled-coil domains; LZ, leucine zipper; ZF, zinc finger. A summary of ATG9A and TBK1 incorporation

into OPTN Fluoppi foci is shown in the rightmost panel. **(B)** Representative microscopic images of ATG9A incorporation into OPTN Fluoppi foci. **(C)** HeLa cells expressing hAG-OPTN and HA-Ash-6Ub were treated with DMSO or BX-795 for 15 h and immunostained with an anti-ATG9A antibody. **(D)** Efficiency of ATG9A incorporation into Fluoppi foci in C. Error bars represent mean ± SD, with 50 Fluoppi foci quantified in two independent experiments. **(E)** HeLa cells expressing hAG-OPTN and HA-Ash-6Ub were treated with DMSO or BX-795 for 15 h and immunostained with an anti-TBK1 (pS172) antibody. **(F)** Efficiency of TBK1 activation in the Fluoppi foci in E. Error bars represent mean ± SD, with 50 Fluoppi foci quantified in two independent experiments. **(G)** HeLa cells transiently expressing 3FLAG-hAG-OPTN and HA-Ash-6Ub were subjected to DSP cross-linking and solubilization with 1% NP-40 and immunoprecipitated with an anti-FLAG antibody. Immunoblots consisted of 5% of input and bound fractions. Red arrowheads denote oligomeric protein bands likely generated by the cross-linking. LDH was used for a negative control. **(H)** Immunoblots of HCT116 WT and TBK1$^{-/-}$ cell lysates. **(I)** WT and TBK1$^{-/-}$ HCT116 cells expressing hAG-OPTN and HA-Ash-6Ub were immunostained. **(J)** Efficiency of ATG9A incorporation into OPTN Fluoppi foci in I. Error bars represent mean ± SD, with 50 Fluoppi foci quantified in two independent experiments. **(K)** Schematic of the FRB-FKBP system for chemically inducing the recruitment of OPTN onto mitochondria. **(L)** HeLa cells transiently expressing OPTN-2FKBP-HA (WT and 4LA) and FRB-FIS1-TM were treated with DMSO or the A/C heterodimerizer for 3 h and then immunostained. **(M)** Quantification of ATG9A recruitment to mitochondria in L. Error bars represent mean ± SD, with >100 cells counted in each of three independent experiments. Scale bars, 10 μm (B, C, E, I, and L). n.s., not significant; ***, P < 0.001.

We also elucidated the molecular function of OPTN, a critical mitophagy adaptor. OPTN and a key regulatory kinase, TBK1, both of which have been identified as genes linked to familial or sporadic amyotrophic lateral sclerosis, interact and are required for efficient mitophagy (Moore and Holzbaur, 2016; Wong and Holzbaur, 2014). Indeed, previous studies of Parkin-mediated mitophagy demonstrated that the OPTN–TBK1 axis promotes phosphorylation of OPTN S473 and S177, which contributes to higher binding affinities for Ub and ATG8, respectively (Heo et al., 2015; Moore and Holzbaur, 2016; Richter et al., 2016; Wild et al., 2011). Consistent with this, Parkin-mediated mitophagy promoted the assembly of OPTN (and NDP52), rather than p62 and NBR1, at the junction between ubiquitinated mitochondria and the autophagosomal formation site. However, by generating liquid phase–separated fluorescent foci in cells, we found that the binding affinities of OPTN (and NDP52) for ATG8 homologues did not exceed those of p62 and NBR1. The binding affinity for ATG8 thus cannot account for the critical roles of OPTN and NDP52 in Parkin-mediated mitophagy, which were also reported recently (Padman et al., 2019; Vargas et al., 2019). To solve this seeming discrepancy, we continuously used the Fluoppi assay and found that OPTN forms a complex with ATG9A vesicles. A single ATG9A vesicle is composed of 30 ATG9A molecules (Yamamoto et al., 2012). Furthermore, multiple ATG8 family proteins are anchored to a single isolation membrane. Therefore, OPTN can interact with multiple ATG8s and ATG9A molecules via the separately coded OPTN–ATG8 and OPTN–ATG9A axes to seed local autophagosomal membrane formation. Consistent with the defect in heterogeneous localization to damaged mitochondria, mitophagy was reduced by an OPTN mutant unable to interact with ATG9A, whereas all mitophagic functions were lost with an OPTN mutant that blocked interactions with ATG8 proteins and ATG9A. Recently, another critical autophagy adaptor, NDP52, was reported to also have binding sites for two autophagy proteins, an NDP52–FIP200 axis and an NDP52–ATG8 axis (Ravenhill et al., 2019; Vargas et al., 2019). This study, in conjugation with previous reports, indicates that the association of OPTN (and NDP52) with ubiquitinated mitochondria promotes the formation of an initial platform that triggers the assembly of different autophagy core units through multivalent interactions. These functionalities thus fulfill a critical role for de novo autophagosomal membrane formation close to the ubiquitinated cargo.

## Materials and methods

Reagents used in this study, including cell lines, antibodies, and plasmid DNAs (including siRNA), are listed in Table 1, Table 2, and Table 3, respectively.

### Chemical synthesis of SNIPER(CRABP)-11

All reagents and solvents were purchased from Sigma-Aldrich, Wako Pure Chemical, or Tokyo Chemical Industry and used without purification. Analytical TLC was conducted using Merck silica gel 60 F254–precoated plates and visualized using a 254-nm UV lamp, phosphomolybdic acid, *p*-anisaldehyde, or ninhydrin stains. Column chromatography was performed using silica gel (spherical, neutral) purchased from Kanto Chemical. $^1$H-NMR spectra were measured using a Varian AS 400 Mercury spectrometer. Chemical shifts are expressed in parts per million downfield from a solvent residual peak or internal standard tetramethylsilane. Mass spectra were measured using a Shimadzu IT-TOF MS equipped with an electrospray ionization source.

1-[Bis(dimethylamino)methylene]-1$H$-benzotriazolium 3-oxide hexafluorophosphate (283.7 mg, 0.75 mmol) was added to a solution of compound 1 (Itoh et al., 2012; 473.4 mg, 0.5 mmol as a HCl salt), compound 2 (Itoh et al., 2010; 223.3 mg, 0.51 mmol), and $N,N$-diisopropylethylamine (259.0 mg, 2 mmol) in acetonitrile (10 ml). The resulting mixture was stirred at room temperature for 3.5 h. The reaction mixture was quenched with water and extracted with AcOEt. The organic layer was washed with brine and then dried over $Na_2SO_4$. After filtration, evaporation of the solvent in vacuo, and purification of the residue by flash column chromatography (4% methanol/$CH_2Cl_2$), compound 3 was obtained as a yellow form (516.6 mg, 77% yield); MS (ESI) $m/z$: 1359 [M + Na$^+$].

1 M tetrabutylammonium fluoride in tetrahydrofuran (2.5 ml, 2.5 mmol) was added to a solution of the compound 3 above (131.9 mg, 0.1 mmol) and MeOH (0.320 ml, 8 mmol) in tetrahydrofuran (5 ml). The reaction mixture was stirred at room temperature for 3 h, then purified by flash column chromatography (10% methanol/CHCl$_3$) to yield 36.7 mg of SNIPER(CRABP)-11 (4; 35%) as a yellow solid; $^1$H-NMR (400 MHz, chloroform-*d*) δ 7.75 – 7.61 (m, 1H), 7.43 – 7.12 (m, 12H), 7.05 – 6.92 (m, 1H), 6.88 – 6.66 (m, 1zH), 6.40 (d, $J$ = 14.8 Hz, 1H), 6.36 – 6.08 (m, 4H), 5.91 (s, 1H), 5.17 (t, $J$ = 9.2 Hz, 1H), 4.63 – 4.33 (m, 4H), 3.79 – 3.65 (m, 1H), 3.63 – 3.25 (m, 13H), 3.22 – 3.04 (m, 1H), 2.90 (br, 1H), 2.83 – 2.42 (m, 4H), 2.42 – 2.25 (m, 8H), 2.03 (s, 3H), 1.98 – 1.45 (m, 12H), 1.46 – 1.14 (m, 7H), 1.09 (s, 6H); high-resolution mass spectrometry

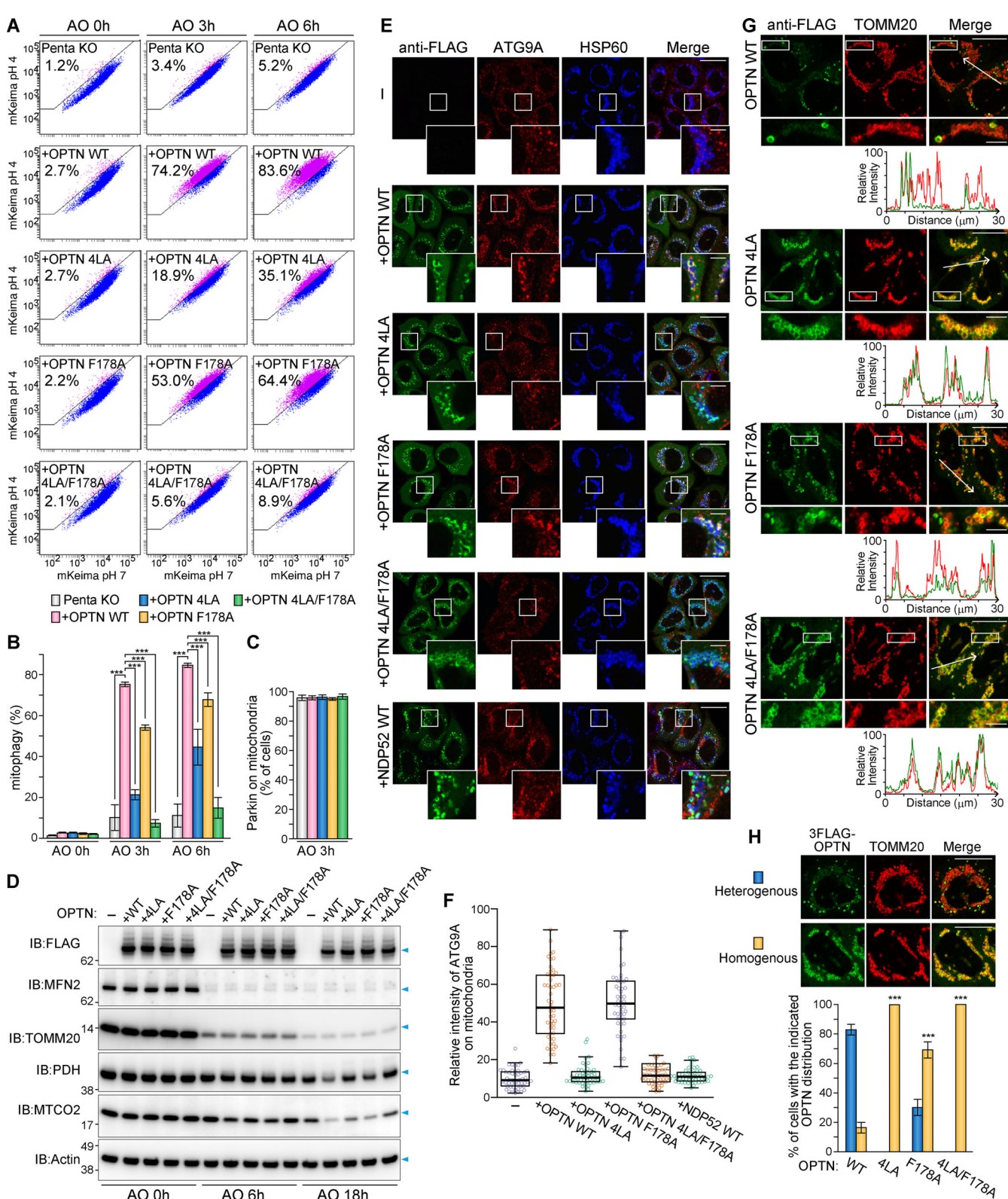

Figure 7. **OPTN–ATG9A interaction is critical for efficient mitophagy. (A)** Penta KO HeLa cells stably expressing YFP-Parkin and mt-Keima with or without the indicated 3FLAG-OPTN were treated with AO and analyzed by FACS. Representative FACS plots with the mt-Keima 561/488-nm ratio are indicated. **(B)** Quantification of FACS-based mitophagy in A. Error bars represent mean ± SD in three independent experiments. **(C)** Quantification of GFP-Parkin translocation to mitochondria in Penta KO HeLa cells with or without the indicated 3FLAG-OPTN after treatment for 3 h with AO. Error bars represent mean ± SD, with >100 cells counted in each of three independent experiments. **(D)** Immunoblots of Penta KO cells stably expressing untagged Parkin with or without the indicated 3FLAG-OPTN following AO treatment. **(E)** Penta KO HeLa cells stably expressing untagged Parkin and the indicated 3FLAG-OPTN or NDP52 were treated with valinomycin for 3 h and then immunostained. HSP60 was used as a mitochondrial marker. Scale bars, 10 μm; insets, 2 μm. **(F)** Quantification of

ATG9A recruitment to mitochondria in cells in E. Error bars represent mean ± SD, with 50 Fluoppi foci quantified in two independent experiments. **(G)** HeLa cells stably expressing untagged Parkin and the indicated 3FLAG-OPTN were treated with valinomycin for 3 h and then immunostained. Line scan plots of the fluorescent signals in the merged images (indicated by the white arrows) are shown in the lower panel. Scale bars, 10 µm; insets, 2 µm. **(H)** Quantification of cells with either a heterogeneous or a homogeneous OPTN distribution on mitochondria after treatment with valinomycin for 3 h. Representative images are shown in the upper panel. Error bars represent mean ± SD, with >100 cells counted in each of three independent experiments. Scale bars, 10 µm. ***, P < 0.001.

(electrospray ionization) calculated for $C_{60}H_{84}N_7O_{10}$ ([M + H]$^+$), 1,062.6274; found 1,062.6278.

## Plasmid construction

### 3FLAG and 3HA tagging

3FLAG-TEV (Tobacco Etch Virus [TEV] protease cleavage site: ENLYFQS), 3HA-TEV, TEV-3FLAG, and TEV-3HA coding sequences were inserted into the BamHI/EcoRI sites of pMXs-puro retrovirus vector to make pMXs-puro_3FLAG-TEV, pMXs-puro_3HA-TEV, pMXs-puro_TEV-3FLAG, and pMXs-puro_TEV-3HA, respectively. Then, 3FLAG-TEV, 3HA-TEV, TEV-3FLAG, and TEV-3HA coding sequences, including 30 bp of the 5′ upstream (5′-CTA GACTGCCGGATCTAGCTAGTTAATTAA-3′) and 30 bp of the 3′ downstream (5′-CAGCTGAGCGCCGGTCGCTACCATTACCAG-3′), were amplified from the above pMXs-puro plasmids and inserted into the BamHI/EcoRI sites of pBABE-puro retrovirus vector or the HindIII/XhoI sites of pcDNA3.1(+) vector to make pBABE-puro_3FLAG-TEV, pBABE-puro_3HA-TEV, pBABE-puro_TEV-3FLAG, pBABE-puro_TEV-3HA, pcDNA3.1(+)_3FLAG-TEV, pcDNA3.1(+)_3HA-TEV, pcDNA3.1(+)_TEV-3FLAG, and pcDNA3.1(+)_TEV-3HA. All of the plasmids contain unique BamHI and EcoRI sites 3′ downstream of the TEV site for 3FLAG-TEV and 3HA-TEV or 5′ upstream for TEV-3FLAG and TEV-3HA, in which a gene of interest can be inserted by the seamless ligation cloning extract (SLiCE) method (Zhang et al., 2014), with minor modifications, or the Gibson Assembly system.

### Expression of linear Ub chains

Plasmids for mammalian expression of linear Ubs were constructed as follows. Human Ub WT, K48R, and K0 were amplified by PCR using the primer pair BamHI-Ser-Ub-F (5′-GGCCGG ATCCTCCATGCAGATTTTCGTG-3′) and Ub(-73)-BglII-Ala-EcoRI-R (5′-GGCCGAATTCTGCAGATCTAAGTCTCAACACAAGATGAA-3′). The amplified Ub gene was treated with BamHI and EcoRI and inserted into the BamHI/EcoRI sites of pBluescriptII SK(−) vector to make pBSK_BamHI-Ser-Ub(1–73, WT, K48R, or K0)-BglII-Ala-EcoRI. To make a double tandem Ub repeat (2Ub), the BamHI/EcoRI-treated Ub gene isolated from the above pBSK plasmid was inserted into the BglII/EcoRI sites of the above pBSK plasmid. Four tandem and six tandem Ub repeats (4Ub and 6Ub) were similarly constructed. The resultant Ub genes have G75S/G76S mutations that block cleavage by deubiquitinating enzymes. The coding sequence for the N-terminal 49 aa of TOMM20 was initially inserted into pcDNA3.1(+)_TEV-3HA to make pcDNA3.1(+)_TOMM20(1–49)-TEV-3HA, and then the YFP gene was inserted to make pcDNA3.1(+)_TOMM20(1–49)-YFP-TEV-3HA. Finally, the respective tandem Ub-chain sequences were inserted into the BamHI/EcoRI sites of pcDNA3.1(+)_TOMM20(1–49)-YFP-TEV-3HA for expression of mitochondria-targeted linear Ub chains.

### Expression of CRABP-II fusion proteins

The CRABP-II coding sequence was amplified by PCR from the plasmid described previously (Okuhira et al., 2017) and inserted into either the BamHI site of pMXs-puro_3HA-TEV vector or the EcoRI site of pMXs-puro_TEV-3HA vector to make pMXs-puro_3HA-TEV-CRABP-II and pMXs-puro_CRABP-II-TEV-3HA, respectively. Then, coding sequences for TOMM20 and HK1 (Okatsu et al., 2012) were inserted into the BamHI/EcoRI sites of pMXs-puro_CRABP-II-TEV-3HA.

### Stable expression of autophagy adaptors

The coding sequences for the human autophagy adaptors (OPTN, NDP52, p62, NBR1, and TAX1BP1) were amplified from plasmids previously reported (Lazarou et al., 2015) and inserted into pMXs-puro_3FLAG-TEV using the SLiCE method.

### Producing Fluoppi foci with autophagy adaptors in mammalian cells

To detect Ash-tagged proteins, a single HA tag was inserted in front of the Ash-coding region in the pAsh-MCL vector by QuickChange site-directed mutagenesis. The resultant vector was termed pHA-Ash-MCL. Next, sequences for linear 6Ub (WT and K0) were extracted from pBSK plasmids (see the section on expression of linear Ub chains) via BamHI/EcoRI digest and inserted into the BamHI/EcoRI sites of pHA-Ash-MCL. The full-length coding sequences of autophagy adaptors (OPTN, NDP52, p62 and NBR1) and truncated OPTN mutants were inserted into the BamHI/NotI sites of phAG-MCL. The OPTN S473E mutation and various alanine substitutions were introduced into phAG_OPTN plasmid by QuickChange site-directed mutagenesis with the appropriate primer pairs. For stable expression of 3FLAG-tagged hAG-OPTN, DNA fragments coding hAG-OPTN were inserted into pMXs-puro_3FLAG-TEV using the SLiCE method.

### FRB-FKBP dimerization assay

The FRB-FIS1 plasmid was described previously (Lazarou et al., 2012). pOPTN(1–444)-2FKBP-HA was constructed by replacing via EcoRI/XbaI digest the PINK1 Δ110-YFP portion of the PINK1Δ110-YFP-FKBP plasmid (Lazarou et al., 2012), which has two tandem FKBP followed by a single HA tag, with PCR-amplified DNA corresponding to OPTN (1–444 aa) with or without a 4LA mutation.

### Other cloning

The coding sequences for human ATG8 proteins (LC3A, LC3B, LC3C, GABARAP, GABARAPL1, and GABARAPL2), human TBK1, and human ATG14 were amplified from plasmids previously reported (Nguyen et al., 2016; Yamano et al., 2014), pWZL Neo Myr Flag TBK1 (a gift from W. Hahn and J. Zhao, Dana-Farber Cancer Institute, Boston, MA), and pMXS-IP GFP-Atg14 (a gift

| Reagent or resource | Source | Identifier |
|---|---|---|
| **Reagents** | | |
| Antimycin A | Sigma-Aldrich | Catalog number A8674 |
| Oligomycin | Calbiochem | Catalog number 495455-10MGCN |
| Valinomycin | Sigma-Aldrich | Catalog number V0627 |
| A/C heterodimerizer | Clontech | Catalog number 635057 |
| Q-VD-OPH | SM Biochemicals | Catalog number SMPH001 |
| Bafilomycin A1 | Calbiochem | Catalog number 196000 |
| SNIPER(CRABP)-11 | This study | N/A |
| DMEM | Sigma-Aldrich | Catalog number D5796 |
| McCoy's 5A | Gibco | Catalog number 16600-082 |
| FBS | Biowest | Catalog number S1820 |
| MEM nonessential amino acids solution | Gibco | Catalog number 11140-050 |
| Sodium pyruvate | Gibco | Catalog number 11360-070 |
| Penicillin–Streptomycin–Glutamine | Gibco | Catalog number 10378-016 |
| GlutaMAX supplement | Gibco | Catalog number 35050-061 |
| Hepes | Gibco | Catalog number 15630-080 |
| Opti-MEM | Gibco | Catalog number 31985-070 |
| Trypsin-EDTA (0.25%) | Gibco | Catalog number 25200-056 |
| Puromycin | Sigma-Aldrich | Catalog number P8833 |
| G418 | Sigma-Aldrich | Catalog number G8168-10ML |
| Hygromycin B | Invitrogen | Catalog number 10687010 |
| EF1-hspCas9-H1-gRNA linearized SmartNuclease vector | System Biosciences | Catalog number CAS900A-1 |
| FuGENE6 | Promega | Catalog number E269A |
| Lipofectamine LTX | Invitrogen | Catalog number 15338100 |
| Lipofectamine RNAiMAX | Invitrogen | Catalog number 13778150 |
| Polybrene | Sigma-Aldrich | Catalog number H9268 |
| Dithiothreitol | Roche | Catalog number 10708984001 |
| CHAPS | Sigma-Aldrich | Catalog number C3023 |
| NP-40 | Nacalai Tesque | Catalog number 25223-75 |
| cOmplete, EDTA-free protease inhibitor cocktail | Roche | Catalog number 11873580001 |
| PhosSTOP, phosphatase inhibitor cocktail | Roche | Catalog number 04906845001 |
| MLN-7243 (E1 inhibitor) | Active Biochem | Catalog number A-1384 |
| BX-795 (TBK1 inhibitor) | Abcam | Catalog number ab142016 |
| NEM | Wako Pure Chemical | Catalog number 058-02061 |
| DDDDK-tagged Protein PURIFICATION GEL | MBL | Catalog number 3329 |
| DSP | Thermo Fisher Scientific | Catalog number 22585 |
| 4% PFA solution | Wako Pure Chemical | Catalog number 163-20145 |
| Gelatin | Sigma-Aldrich | Catalog number G9391 |
| Triton X-100 | MP Biomedicals | Catalog number 807426 |
| Digitonin | Wako Pure Chemical | Catalog number 043-21371 |
| Tween 20 | Nacalai Tesque | Catalog number 35624-15 |
| NuPAGE 4–12% Bis-Tris gel | Invitrogen | Catalog number NP0323BOX |
| NuPAGE MOPS SDS running buffer | Invitrogen | Catalog number NP0001 |
| NuPAGE MOPS SDS running buffer | Invitrogen | Catalog number NP0002 |

Table 1. **Reagents and cell lines used in this study (Continued)**

| Reagent or resource | Source | Identifier |
| --- | --- | --- |
| Difco skim milk | BD Biosciences | Catalog number 232100 |
| BSA (fatty acid free) | Sigma-Aldrich | Catalog number A8806 |
| Phusion high-fidelity DNA polymerase | Thermo Fisher Scientific | Catalog number F-530L |
| DNA ligation kit | TaKaRa | Catalog number 6023 |
| Western Lightning Plus-ECL | PerkinElmer | Catalog number NEL105001EA |
| ECL Prime Western blotting detection reagents | GE Healthcare | Catalog number RPN2232 |
| **Cell lines** | | |
| HeLa (corresponds to WT for PINK1 KO HeLa) | Okatsu et al., 2015 | |
| PINK1 KO HeLa | Okatsu et al., 2015 | |
| HeLa (corresponds to WT for FIP200, ATG5, ATG9A, Penta KO HeLa) | ATCC | CCL-2,2 |
| FIP200 KO HeLa | Vargas et al., 2019 | |
| ATG5 KO HeLa | Nezich et al., 2015 | |
| ATG9A KO HeLa | Nezich et al., 2015 | |
| Penta (OPTN/NDP52/TAX1BP1/p62/NBR1) KO HeLa | Lazarou et al., 2015 | |
| HEK293T | ATCC | CRL-3216 |
| HT1080 | Itoh et al., 2012 | |
| HCT116 | Yamano et al., 2018 | |
| TBK1$^{-/-}$ HCT116 | This study | |

from N. Mizushima, The University of Tokyo, Tokyo, Japan), respectively. Products were inserted into pMXs-puro_3FLAG-TEV vector. To stably express untagged Parkin, the human PARK2 coding sequence was subcloned into the pBABE-puro vector.

### Cell culture and transfection

HeLa, HEK293T, and HT1080 cell lines were cultured in DMEM supplemented with 10% (vol/vol) FBS, 1 mM sodium pyruvate, nonessential amino acids, and penicillin–streptomycin–glutamine (PSG) or 2 mM GlutaMax. FIP200 KO HeLa (Vargas et al., 2019), ATG5 KO HeLa (Nezich et al., 2015), ATG9 KO HeLa (Nezich et al., 2015), Penta KO HeLa (Lazarou et al., 2015), and PINK1 KO HeLa (Okatsu et al., 2015) cell lines were described previously. HCT116 WT and TBK1$^{-/-}$ cell lines were cultured in McCoy's 5A medium supplemented with 10% (vol/vol) FBS and 2 mM GlutaMax. Cells were cultured at 37°C in a 5% $CO_2$ incubator.

Stable cell lines were established by recombinant retrovirus infection as follows. First, Gag-Pol, VSV-G, and a retroviral plasmid were cotransfected into HEK293T cells grown in a 6-well plate using Lipofectamine LTX. At 12 h after transfection, the respective culture media were changed to fresh media supplemented with Hepes, and the cells cultivated for 24 h. Filtrated viral supernatants were infected into HeLa or HT1080 cells with 8 µg/ml polybrene. FuGENE6 transfection reagent was used according to the manufacturer's instructions for transient expression.

Valinomycin was used at a final concentration of 10 µM to induce robust Parkin translocation. Oligomycin and antimycin A

were used at final concentrations of 10 µM and 4 µM, respectively, to induce mitophagy without lysosomal neutralization. Bafilomycin A1 was used at a final concentration of 100 nM to inhibit lysosomal acidification. When cells were treated with oligomycin and antimycin A for >6 h, 10 µM Q-VD-OPH was added to block apoptotic cell death. SNIPER(CRABP)-11 dissolved in DMSO was used at a final concentration of 1 µM. The A/C heterodimerizer and BX-795 were used at final concentrations of 0.5 µM and 2 µM, respectively.

### Construction of TBK1$^{-/-}$ HCT116 cell line

TBK1$^{-/-}$ HCT116 cell lines were established by CRISPR/Cas9-based genome editing with an antibiotics-selection strategy. The gRNA target sequence for exon 6 in the TBK1 gene (5′-GGCCCT TCAAAGGGTCTAAATGG-3′) was selected using CRISPOR, an online gRNA design tool (http://crispor.tefor.net/). Two DNA oligonucleotides, TBK1-ex6-CRISPR-F3 (5′-TGTATGAGACCACG GCCCTTCAAAGGGTCTAAA-3′) and TBK1-ex6-CRISPR-R3 (5′-AAACTTTAGACCCTTTGAAGGGCCGTGGTCTCA-3′), were annealed and introduced into a linearized pEF1-hspCas9-H1-gRNA vector according to the manufacturer's instructions. Neomycin-resistant (NeoR) and hygromycin resistant (HygroR) marker plasmids were constructed as follows. Portions (247 bp) of the 5′ and 3′ homology arms of TBK1 exon 6, which lacks the gRNA sequence but has a BamHI site in the middle (total 500 bp), were synthesized and cloned into a pUC-Amp vector (GENEWIZ) to make pUC57-Amp_TBK1-ex6-donor. NeoR and HygroR DNA fragments extracted by BamHI from pBSK_NeoR and pBSK_HygroR (Koyano et al., 2019a) were inserted into

Table 2.  **Antibodies used in this study**

| Reagent or resource | Source | Identifier |
| --- | --- | --- |
| **Antibodies** | | |
| Mouse monoclonal anti-HA (TANA2) | MBL | Catalog number M180-3, RRID:AB_10951811 |
| Rabbit polyclonal anti-TOMM20 (FL-145) | Santa Cruz Biotechnology | Catalog number sc-11415, RRID:AB_2207533 |
| Mouse monoclonal anti-MTCO2 (12C4F12) | Abcam | Catalog number Ab110258 |
| Mouse monoclonal anti-actin (C4) | Merck Millipore | Catalog number MAB1501R, RRID:AB_2223041 |
| Mouse monoclonal anti-DDDDK (FLA-1) | MBL | Catalog number M185-3L, RRID:AB_11123930 |
| Rabbit polyclonal anti-LC3B | Sigma-Aldrich | Catalog number L7543, RRID:AB_796155 |
| Guinea pig polyclonal anti-p62 | MBL | Catalog number PM066, RRID:AB_10952738 |
| Rabbit polyclonal anti-LC3B for immunofluorescence | MBL | Catalog number PM036, RRID:AB_2274121 |
| Mouse monoclonal anti-multi Ub (FK2) | MBL | Catalog number D058-3, RRID:AB_592937 |
| Rabbit monoclonal anti-Ub, Lys48-specific (Apu2) | Merck Millipore | Catalog number 05-1307, RRID:AB_1587578 |
| Rabbit monoclonal anti-Ub, Lys63-specific (Apu3) | Merck Millipore | Catalog number 05-1308, RRID:AB_1587580 |
| Mouse monoclonal anti-TOMM22 (1C9-2) | Sigma-Aldrich | Catalog number T6319, RRID:AB_261724 |
| Rabbit polyclonal anti-Azami-Green | MBL | Catalog number PM011M |
| rabbit polyclonal anti-TBK1 | Cell Signaling Technology | Catalog number 3013S, RRID:AB_2199749 |
| Rabbit monoclonal anti-phospho TBK1 (S172; D52C2) | Cell Signaling Technology | Catalog number 5483, RRID:AB_10693472 |
| Rat monoclonal anti-HA (3F10) | Roche | Catalog number 11867423001, RRID:AB_390918 |
| Rabbit polyclonal anti-DDDDK | MBL | Catalog number PM020, RRID:AB_591224 |
| Rabbit monoclonal anti-LDH (EP1566Y) | Abcam | Catalog number ab52488, RRID:AB_2134961 |
| Rabbit monoclonal anti-ATG13 (E1Y9V) | Cell Signaling Technology | Catalog number 13468S, RRID:AB_2797419 |
| Rabbit polyclonal anti-WIPI2 | Sigma-Aldrich | Catalog number SAB4200400 |
| rabbit polyclonal anti-ATG16L1 | MBL | Catalog number PM040, RRID:AB_1278757 |
| Rabbit monoclonal anti-ATG9A (EPR2450(2)) | Abcam | Catalog number ab108338, RRID:AB_10863880 |
| Rabbit monoclonal anti-FIP200 (D10D11) | Cell Signaling Technology | Catalog number 12436S, RRID:AB_2797913 |
| Rabbit polyclonal anti-OPTN | Proteintech | Catalog number 10837-1-AP, RRID:AB_2156665 |
| Rabbit monoclonal anti-NDP52 (D1E4A) | Cell Signaling Technology | Catalog number 60732, RRID:AB_2732810 |
| Mouse monoclonal anti-TOMM20 (F-10) | Santa Cruz Biotechnology | Catalog number sc-17764, RRID:AB_628381 |
| Mouse monoclonal anti-MFN2 (6A8) | Abcam | Catalog number ab56889, RRID:AB_2142629 |
| Mouse monoclonal anti-PDHA1 (8D10E6) | Abcam | Catalog number ab110334, RRID:AB_10866116 |
| Mouse monoclonal anti-PINK1 (38CT18.7) | LSBio | Catalog number LS-C96472, RRID:AB_10559463 |
| Goat polyclonal anti-HSP60 (N-20) | Santa Cruz Biotechnology | Catalog number sc-1052, RRID:AB_631683 |
| Goat anti-rabbit IgG Alexa Fluor 488 conjugated | Invitrogen | Catalog number A-11034 |
| Goat anti-rabbit IgG Alexa Fluor 568 conjugated | Invitrogen | Catalog number A-11036 |
| Goat anti-mouse IgG Alexa Fluor 488 conjugated | Invitrogen | Catalog number A-11029 |
| Goat anti-mouse IgG Alexa Fluor 568 conjugated | Invitrogen | Catalog number A-11031 |
| Goat anti-mouse IgG Alexa Fluor 647 conjugated | Invitrogen | Catalog number A-21236 |
| Goat anti-rat IgG Alexa Fluor 488 conjugated | Invitrogen | Catalog number A-11006 |
| Goat anti-rat IgG Alexa Fluor 568 conjugated | Invitrogen | Catalog number A-11077 |
| Goat anti-rat IgG Alexa Fluor 647 conjugated | Invitrogen | Catalog number A-21247 |
| Goat anti-guinea pig IgG Alexa Fluor 568 conjugated | Invitrogen | Catalog number A-11075 |
| Donkey anti-mouse IgG Alexa Fluor 488 conjugated | Abcam | Catalog number ab150105 |
| Donkey anti-rabbit IgG Alexa Fluor 568 conjugated | Abcam | Catalog number ab175470 |
| Donkey anti-goat IgG Alexa Fluor 647 conjugated | Invitrogen | Catalog number A-21447 |
| Goat Anti-rabbit IgG horseradish peroxidase-linked | Jackson ImmunoResearch | Catalog number 111-035-144 |
| Anti-mouse IgG horseradish peroxidase-linked | Promega | Catalog number W402B |

the BamHI site of pUC57-Amp_TBK1-ex6-donor to make pUC57-Amp_TBK1-ex6-loxP-NeoR and pUC57-Amp_TBK1-ex6-loxP-HygroR, respectively. The resultant NeoR and HygroR plasmids were transfected into HCT116 cells with the gRNA plasmid using FuGENE6. Cells were grown in McCoy's 5A media containing 700 µg/ml G418 and 100 µg/ml hygromycin B. Single clones were then isolated and screened by PCR using genomic DNA to verify insertions of NeoR and HygroR into exon 6 of the TBK1 gene. Finally, TBK1 gene depletion was confirmed by immunoblotting.

### RNA interference

siRNA oligonucleotides for PINK1 and a nontargeting control were described previously (Koyano et al., 2014). siRNAs were transfected into cells using Lipofectamine RNAiMAX according to the manufacturer's instructions. At 24 h after transfection, the medium was changed and the cells were grown for another 24 h before analysis.

### Live-cell and immunofluorescence microscopy

Live-cell imaging samples were prepared by culturing cells on glass-bottom 35-mm dishes (MatTek). For immunofluorescent imaging samples, cells grown on glass-bottom 35-mm dishes (MatTek) were fixed with 4% PFA in PBS for 25 min at room temperature. When detecting ubiquitinated proteins with anti-Apu2 or anti-Apu3 antibodies, methanol fixation was performed. Ice-cold methanol was added to cells for 15 min at −20°C followed by three PBS washes. The cells were permeabilized with 0.15% (vol/vol) Triton X-100 or 50 µg/ml digitonin (for observing LC3B) in PBS for 20 min and preincubated with 0.1% (wt/vol) gelatin in PBS for 30 min. The cells were then incubated with primary antibodies diluted in 0.1% (wt/vol) gelatin in PBS for 2 h at room temperature, washed three times with PBS with Tween 20 (PBS-T), and then incubated for 1 h with the appropriate secondary antibodies conjugated to Alexa Fluor. Microscopy images were captured using an inverted confocal microscope (LSM780; Carl Zeiss) with a Plan-Apochromat 63×/1.4 oil differential interference contrast lens. For image analysis, ZEN microscope software and Photoshop (Adobe) were used. For live-cell imaging, a fluorescence signal for mt-Keima in neutral pH was obtained following excitation with a 458-nm laser (green), whereas the fluorescence signal in acidic pH (i.e., in lysosomes) was obtained following excitation with a 561-nm laser (red). As Keima has a single emission peak at 620 nm, a 600–720-nm emission range was used for both the 458- and 561-nm excitations. The relative intensities of ATG8s or ATG9A in Fluoppi foci were calculated using ImageJ. In brief, the mean fluorescent values of immunostained ATG8s or ATG9A in a Fluoppi spot and those in cytoplasmically localized in the same cell were measured, with the ratio of the Fluoppi spot signal to the cytosolic signal defined as relative intensity.

### Immunoprecipitation

3FLAG-hAG-OPTN and HA-Ash-6Ub were transiently expressed in HeLa cells grown in a 6-cm dish. Cells were washed twice with PBS, treated with 1 mM dithiobis(succinimidyl propionate) (DSP) in 1 ml PBS for 30 min at room temperature by gentle rocking and then treated for another 10 min with 200 mM

Tris-HCl, pH 7.4, for quenching. After a TBS buffer wash, the cells were solubilized on ice for 20 min with 1% NP-40 buffer (20 mM Tris-HCl, pH 8.0, 150 mM NaCl, 1 mM EDTA, and 1% [vol/vol] NP-40) containing a protease inhibitor cocktail, a phosphatase inhibitor cocktail, 10 µM MLN-7243, and 1 mM N-ethylmaleimide (NEM). After sonication, unsolubilized aggregates were removed by centrifugation at 3,000 g for 5 min at 4°C, and supernatants were incubated with an anti-DDDDK-tag gel for 90 min at 4°C. The gel was washed with 0.1% NP-40 buffer three times, and bound proteins were eluted with SDS-PAGE sample buffer supplemented with DTT.

### Immunoblotting

Cells grown in 6-well plates were washed twice with PBS and solubilized with 2% CHAPS buffer (25 mM Hepes-KOH, pH 7.5, 300 mM NaCl, 2% [wt/vol] CHAPS, and protease inhibitor cocktail) on ice for 20 min. When ubiquitinated proteins were detected, 1 mM NEM was added to the 2% CHAPS buffer. After centrifugation at 12,000 g for 2 min at 4°C, supernatants were collected and protein concentrations determined by Nano-drop analysis. SDS-PAGE sample buffer supplemented with DTT was then added to the supernatants. Alternatively, cells grown in 6-well plates were washed twice with PBS and then simply lysed with appropriate volumes of the SDS-PAGE sample buffer supplemented with DTT followed by sonication to physically shred the genomic DNA. The samples were boiled at 95°C for 5 min. For ATG9A detection, the samples were incubated at 37°C for 30 min rather than 95°C for 5 min. Cell lysates (10–50 µg per lane) were loaded on NuPAGE 4–12% Bis-Tris gels and electrophoresed using MES or MOPS running buffer according to the manufacturer's instructions. Proteins were then transferred to polyvinylidene difluoride membranes, which were blocked with 2% (wt/vol) skim milk/TBS with Tween 20 (TBS-T) for 20 min at room temperature, incubated with primary antibodies diluted in 1% (wt/vol) BSA/TBS-T for 2 h, washed three times with TBS-T, and then incubated with anti-rabbit IgG or anti-mouse IgG horseradish peroxidase-linked secondary antibodies diluted in 2% (wt/vol) skim milk/TBS-T for 1hr. Proteins were detected using a Western Lightning Plus-ECL Kit (and ECL Prime Western Blotting Detection Reagents for PINK1) and an ImageQuant LAS 4000 imaging system (GE Healthcare). ImageJ was used for quantification of protein bands.

### Mitophagy assay using mt-Keima and FACS

When Parkin-mediated mitophagy was measured, HeLa cells stably expressing YFP-Parkin and mt-Keima grown in a 6-well plate were treated with AO for the indicated time periods. Cells were washed once with PBS and then detached with 0.05% trypsin EDTA. When OMM-Ub–induced mitophagy was measured, YFP-tagged OMM-targeted linear Ub chains were transiently expressed in HeLa cells stably expressing mt-Keima for 48 h. When SNIPER-induced mitophagy was measured, HT1080 cells stably expressing mt-Keima and cytosolic YFP were treated with 1 µM SNIPER(CRABP)-11 for 9 h. Cells were then washed once with PBS and detached with 0.05% trypsin EDTA. Cell pellets after centrifugation were resuspended in sorting buffer (PBS containing 2.5% FBS). Analysis was performed using

Table 3.  **Plasmid DNAs and siRNA used in this study**

| Reagent or resource | Source |
| --- | --- |
| Plasmid DNA | |
| pMXs-puro retroviral vector | Cell Biolabs Inc. (RTV-012) |
| pBABE-puro retroviral vector | Addgene (#1764) |
| pcDNA3.1(+) | Invitrogen (V79020) |
| pBluescriptII SK(-) | Stratagene |
| pMXs-puro_YFP | This study |
| pMXs-puro_3FLAG-TEV | This study |
| pMXs-puro_3HA-TEV | This study |
| pMXs-puro_TEV-3FLAG | This study |
| pMXs-puro_TEV-3HA | This study |
| pBABE-puro_3FLAG-TEV | This study |
| pBABE-puro_3HA-TEV | This study |
| pBABE-puro_TEV-3FLAG | This study |
| pBABE-puro_TEV-3HA | This study |
| pcDNA3.1(+)_3FLAG-TEV | This study |
| pcDNA3.1(+)_3HA-TEV | This study |
| pcDNA3.1(+)_TEV-3FLAG | This study |
| pcDNA3.1(+)_TEV-3HA | This study |
| pcDNA3.1(+)_T20(1-49)-YFP-2Ub(WT)-3HA | This study |
| pcDNA3.1(+)_T20(1-49)-YFP-2Ub(K48R)-3HA | This study |
| pcDNA3.1(+)_T20(1-49)-YFP-2Ub(K0)-3HA | This study |
| pcDNA3.1(+)_T20(1-49)-YFP-4Ub(WT)-3HA | This study |
| pcDNA3.1(+)_T20(1-49)-YFP-4Ub(K48R)-3HA | This study |
| pcDNA3.1(+)_T20(1-49)-YFP-4Ub(K0)-3HA | This study |
| pcDNA3.1(+)_T20(1-49)-YFP-6Ub(WT)-3HA | This study |
| pcDNA3.1(+)_T20(1-49)-YFP-6Ub(K48R)-3HA | This study |
| pcDNA3.1(+)_T20(1-49)-YFP-6Ub(K0)-3HA | This study |
| pMXs-puro_3FLAG-OPTN | This study |
| pMXs-puro_3FLAG-NDP52 | This study |
| pMXs-puro_3FLAG-p62 | This study |
| pMXs-puro_3FLAG-NBR1 | This study |
| pAsh-MCL | MBL (AM-8011M) |
| pHA-Ash-MCL | This study |
| pHA-Ash_6Ub | This study |
| pHA-Ash_6Ub(K0) | This study |
| phAG-MCL | MBL (AM-8011M) |
| phAG_OPTN | This study |
| phAG_OPTN (S177D) | This study |
| phAG_NDP52 | This study |
| phAG_p62 | This study |
| phAG_NBR1 | This study |
| pEYFP-C1 | Clontech (6006-1) |
| pUMVC (Gag-Pol) | Gift from Chunxin Wang |
| pCMV-VSV-G (VSV-G) | Gift from Chunxin Wang |

| Reagent or resource | Source |
| --- | --- |
| pCHAC/EYFP-LC3B-IRES-MCS2 | Yamano et al., 2014 |
| pMXs-puro_3FLAG-LC3A | This study |
| pMXs-puro_3FLAG-LC3B | This study |
| pMXs-puro_3FLAG-LC3C | This study |
| pMXs-puro_3FLAG-GABARAP | This study |
| pMXs-puro_3FLAG-GABARAPL1 | This study |
| pMXs-puro_3FLAG-GABARAPL2 | This study |
| pWZL Neo Myr Flag TBK1 | Addgene (#20648) |
| pMXs-puro_3FLAG-TBK1 | This study |
| pMXS-IP GFP-Atg14 | Addgene (#38264) |
| pMXs-puro_3FLAG-ATG14 | This study |
| phAG_OPTN(S473E) | This study |
| phAG_OPTN(128-577, S473E) | This study |
| phAG_OPTN(150-577, S473E) | This study |
| phAG_OPTN(170-577, S473E) | This study |
| phAG_OPTN(190-577, S473E) | This study |
| phAG_OPTN(210-577, S473E) | This study |
| phAG_OPTN(301-577, S473E) | This study |
| phAG_OPTN(401-577, S473E) | This study |
| phAG_OPTN(1-530, S473E) | This study |
| phAG_OPTN(128-5A-132, S473E) | This study |
| phAG_OPTN(133-5A-137, S473E) | This study |
| phAG_OPTN(138-5A-142, S473E) | This study |
| phAG_OPTN(143-5A-147, S473E) | This study |
| phAG_OPTN(148-5A-152, S473E) | This study |
| phAG_OPTN(L143A, S473E) | This study |
| phAG_OPTN(L150A, S473E) | This study |
| phAG_OPTN(L157A, S473E) | This study |
| phAG_OPTN(L164A, S473E) | This study |
| phAG_OPTN(4LA, S473E) | This study |
| phAG_OPTN(M44Q/L54Q, S473E) | This study |
| pcDNA3.1(+)_3FLAG-hAG-OPTN(S473E) | This study |
| pcDNA3.1(+)_3FLAG-hAG-OPTN(M44Q/L54Q, S473E) | This study |
| pcDNA3.1(+)_3FLAG-hAG-OPTN(4LA, S473E) | This study |
| pEF1-hspCas9-H1-gRNA TBK1-ex6-3 | This study |
| pUC57-Amp_TBK1-ex6 donor | This study |
| pUC57-Amp_TBK1-ex6 loxP-NeoR-donor | This study |
| pUC57-Amp_TBK1-ex6 loxP-HygroR-donor | This study |
| FRB-Fis1 | Lazarou et al., 2012 |
| PINK1Δ110-YFP-FKBP | Lazarou et al., 2012 |
| pOPTN(1-444)-2FKBP-HA | This study |
| pOPTN(1-444, 4LA)-2FKBP-HA | This study |
| pBMNz-YFP-Parkin | Lazarou et al., 2015 |
| pRetroQ-mt-Keima | Gift from Chunxin Wang |

**Table 3.   Plasmid DNAs and siRNA used in this study (*Continued*)**

| Reagent or resource | Source |
|---|---|
| pBABE-puro_Untag-Parkin | This study |
| pMXs-puro_GFP-Parkin | Matsuda et al., 2010 |
| pMXs-puro_3FLAG-OPTN(4LA) | This study |
| pMXs-puro_3FLAG-OPTN(F178A) | This study |
| pMXs-puro_3FLAG-OPTN(4LA, F178A) | This study |
| p3xFLAG-CMV-10_CRABP-II | Okuhira et al., 2017 |
| pMXs-puro_3HA-TEV-CRABP-II | This study |
| pMXs-puro_CRABP-II-TEV-3HA | This study |
| pcDNA3.1-HA_HK1 | Okatsu et al., 2012 |
| pMXs-puro_HK1-CRABP-II-TEV-3HA | This study |
| pMXs-puro_TOMM20-CRABP-II-TEV-3HA | This study |
| Oligonucleotides | |
| siRNA PINK1 (GGGUCAGCACGUUCAGUUAdTdT) | Sigma-Aldrich |

FACSDiva software on a BD LSRFortessa X-20 cell sorter (BD Biosciences). Keima fluorescence was measured using dual-excitation ratiometric pH measurements with 405-nm (pH 7) and 561-nm (pH 4) lasers and 610/20-nm emission filters. For each sample, 10,000 YFP/mt-Keima double-positive cells were collected.

### Statistical analysis
Statistical analysis was performed using data obtained from three or more biologically independent experimental replicates. A $t$ test was used for statistical comparisons between two groups (n.s., not significant; *, $P < 0.05$; **, $P < 0.01$; ***, $P < 0.001$). The sample size ($n$) and number of independent replicates for each experiment are indicated in the figure legends.

### Online supplemental material
Fig. S1 confirms mitochondrial localizations of OMM-targeted Ub chains and provides raw data of Keima-FACS analysis. Fig. S2 confirms PINK1 knockdown in HT1080 cells. Fig. S3 contains microscopic images showing ATG8 recruitment to the autophagy adaptor Fluoppi foci. Fig. S4 examines ATG9A and ATG13 incorporation into Fluoppi foci composed of hAG-OPTN and hAG-NDP52, respectively. Fig. S5 contains microscopic images showing ATG9A recruitment to Fluoppi foci composed of hAG-OPTN WT and mutants and immunoblots of TBK1 activated by OPTN WT and mutants.

## Acknowledgments
We thank Dr. Richard J. Youle and Dr. Chunxin Wang (National Institutes of Health, Bethesda, MD) for the FIP200 KO, ATG5 KO, ATG9A KO, and Penta KO HeLa cells and Dr. Kosuke Tanegashima for FACS analysis.

This work was supported by the Japan Society for the Promotion of Science (KAKENHI grants JP18H05500 and JP18K06237 to K. Yamano; JP17J03737 to W. Kojima; JP18K14708 to F. Koyano; JP18H02443 and JP19H05712 to N. Matsuda; JP19H00997 to K. Tanaka; JP17K08385 to Y. Demizu; and JP18H05502 and JP16H05090 to M. Naito). N. Matsuda was supported by the Chieko Iwanaga Fund for Parkinson's Disease Research and the Takeda Science Foundation. K. Tanaka was supported by the Takeda Science Foundation.

The authors declare no competing financial interests.

Author contributions: K. Yamano, R. Kikuchi, W. Kojima, R. Hayashida, F. Koyano, and J. Kawawaki prepared plasmid materials, constructed stable cell lines, performed the experiments, and analyzed the data. T. Shoda and Y. Demizu synthesized chemical compounds and analyzed the data. K. Yamano, Y. Demizu, M. Naito, K. Tanaka, and N. Matsuda designed the research. K. Yamano wrote the manuscript with help and supervision from K. Tanaka and N. Matsuda.

Submitted: 5 January 2020

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

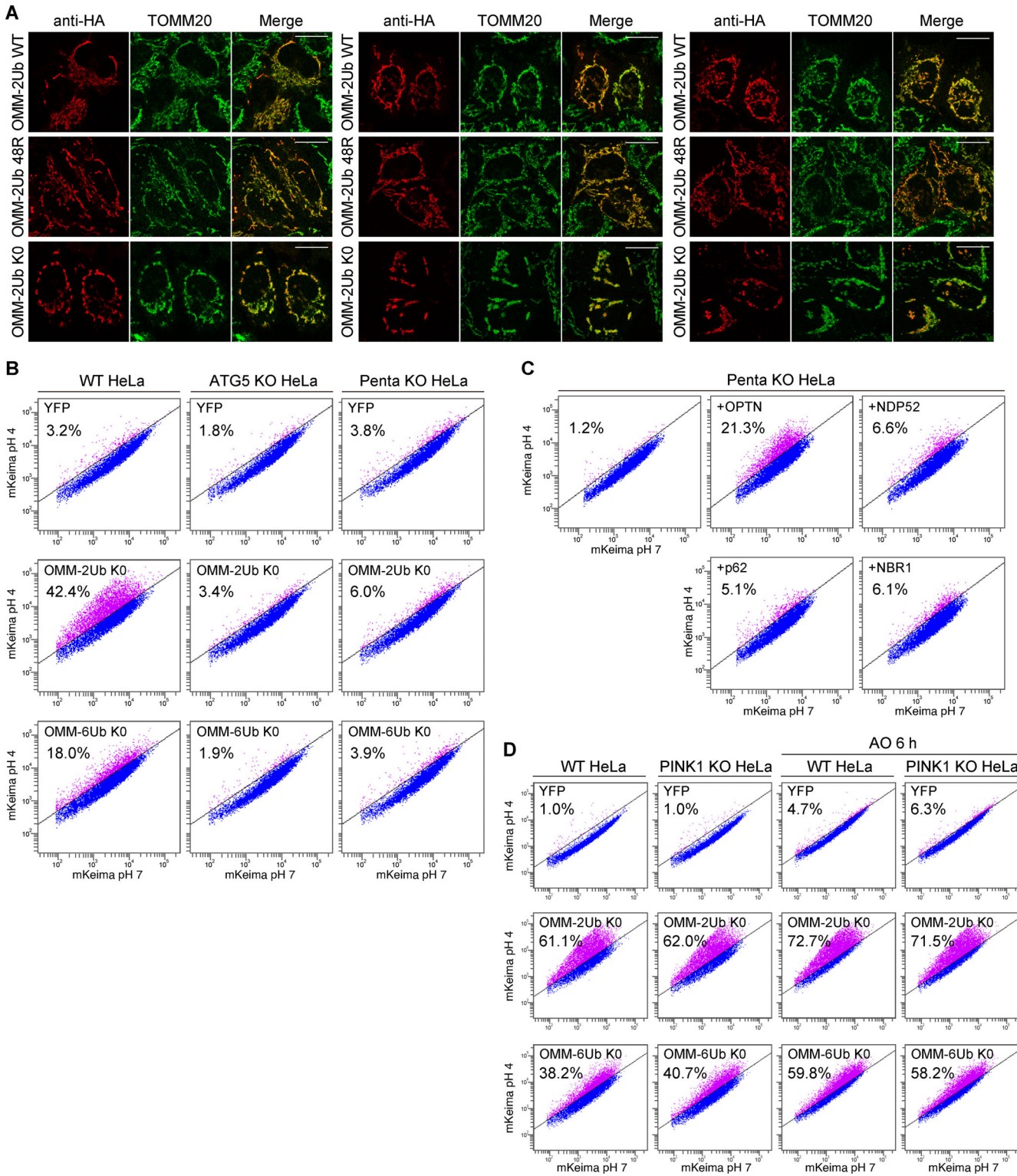

Figure S1. **Ectopic mitochondria-targeted linear Ub chains induce mitophagy. (A)** Mitochondrial localization of linear Ub chains. The indicated OMM-Ub proteins were transiently expressed in HeLa cells. The cells were immunostained with anti-HA and anti-TOMM20 antibodies. Scale bars, 10 μm. **(B)** Cytosolic YFP, OMM-2Ub K0, and OMM-6Ub K0 were transiently expressed in WT, ATG5 KO, or Penta KO HeLa cells stably expressing mt-Keima. At 48 h after transfection, the cells were analyzed by FACS. Representative FACS data (mt-Keima 561/488-nm ratio) with the percentage of cells exhibiting lysosomal positive mt-Keima are shown. **(C)** OMM-2Ub K0 was transiently expressed in Penta KO HeLa cells with stable expressions of each 3FLAG-autophagy adaptor and mt-Keima. At 48 h after transfection, the cells were analyzed by FACS. Representative FACS data (mt-Keima 561/488-nm ratio) with the percentage of cells exhibiting lysosomal positive mt-Keima are shown. **(D)** Cytosolic YFP and the indicated OMM-Ub proteins were transiently expressed in WT and PINK1 KO HeLa cells stably expressing mt-Keima. At 42 h after transfection, the cells were treated with or without AO for 6 h and then analyzed by FACS. Representative FACS data with the percentage of cells exhibiting lysosomal positive mt-Keima are shown.

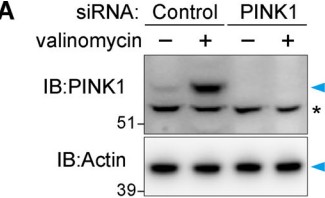

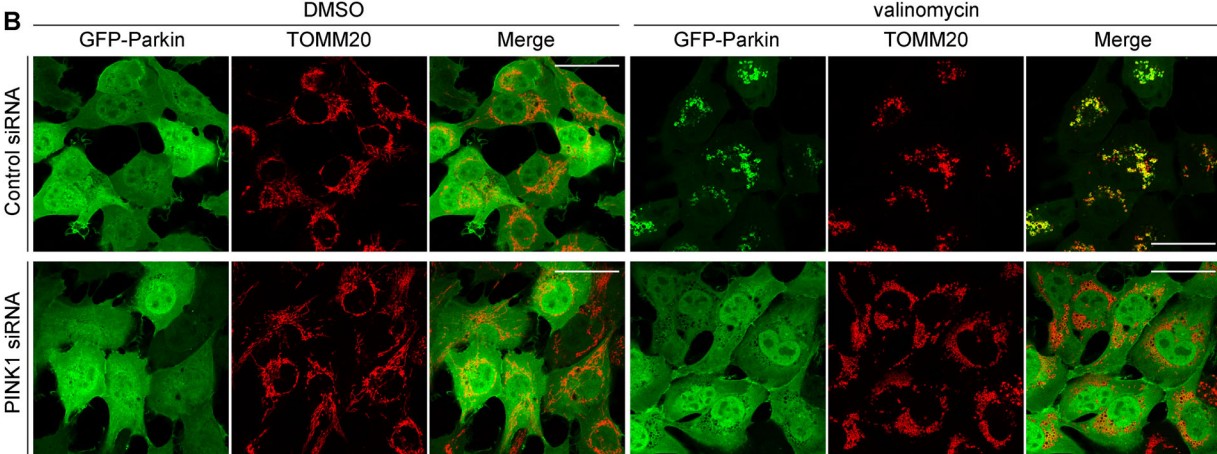

Figure S2. **Validation of PINK1 knockdown in HT1080. (A)** HT1080 cells were treated with control or PINK1 siRNAs for 48 h were then treated with valinomycin for 3 h. Total cell lysates were immunoblotted with anti-PINK1 and anti-actin antibodies. The asterisk denotes nonspecific bands. **(B)** HT1080 cells stably expressing GFP-Parkin treated with control or PINK1 siRNA for 48 h were then treated with valinomycin for 3 h. The cells were immunostained with anti-TOMM20 antibody. Scale bars, 10 μm.

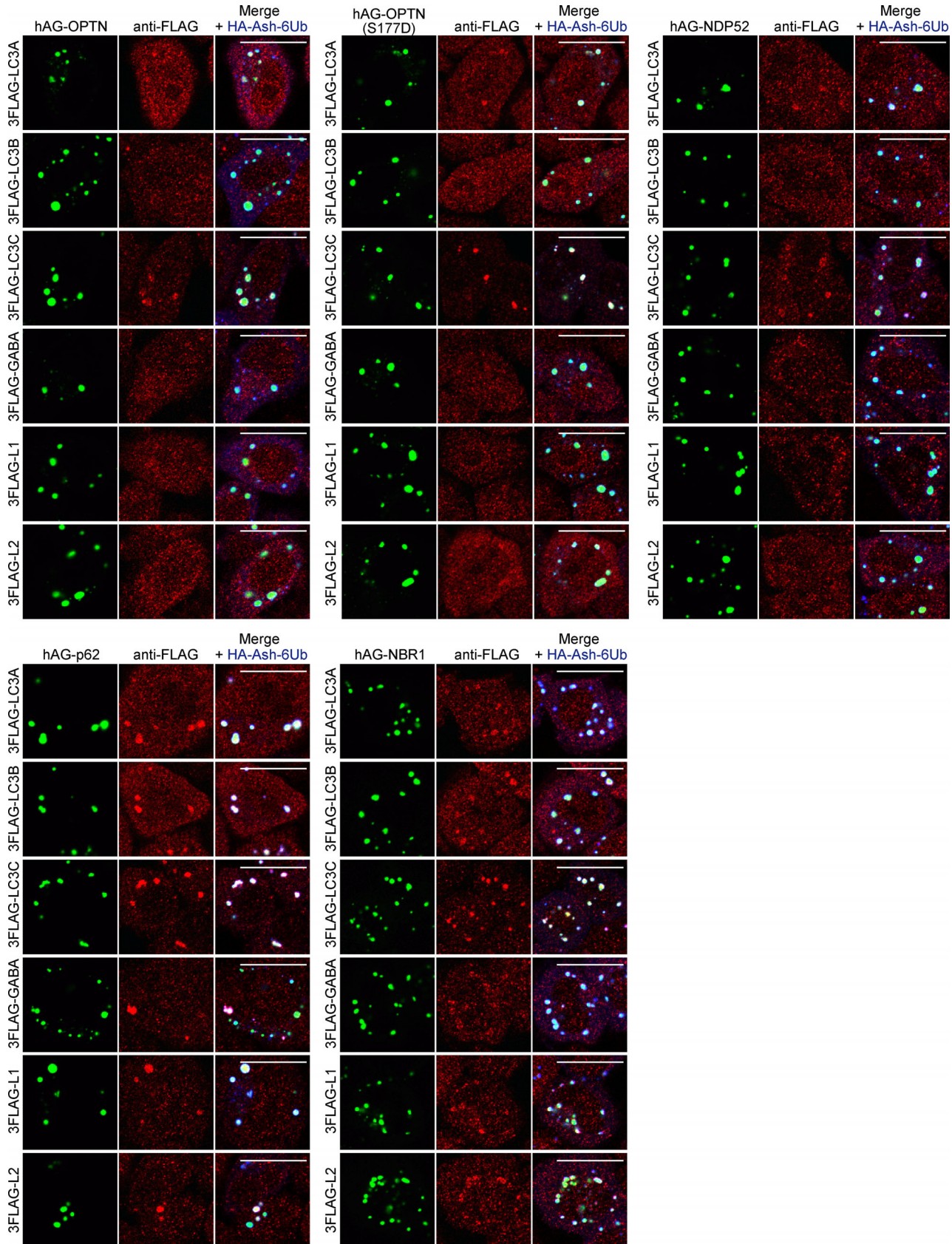

**Figure S3. Incorporation of ATG8 proteins into the Fluoppi foci of autophagy adaptors.** HA-Ash-6Ub– and hAG-tagged autophagy adaptors (OPTN, OPTN(S177D), NDP52, p62, or NBR1) were transiently expressed in ATG5 KO HeLa cells stably expressing 3FLAG-tagged LC3A, LC3B, LC3C, GABARAP (GABA), GABARAPL1 (L1), or GABARAPL2 (L2). The cells were immunostained with anti-FLAG and anti-HA antibodies. Scale bars, 10 µm.

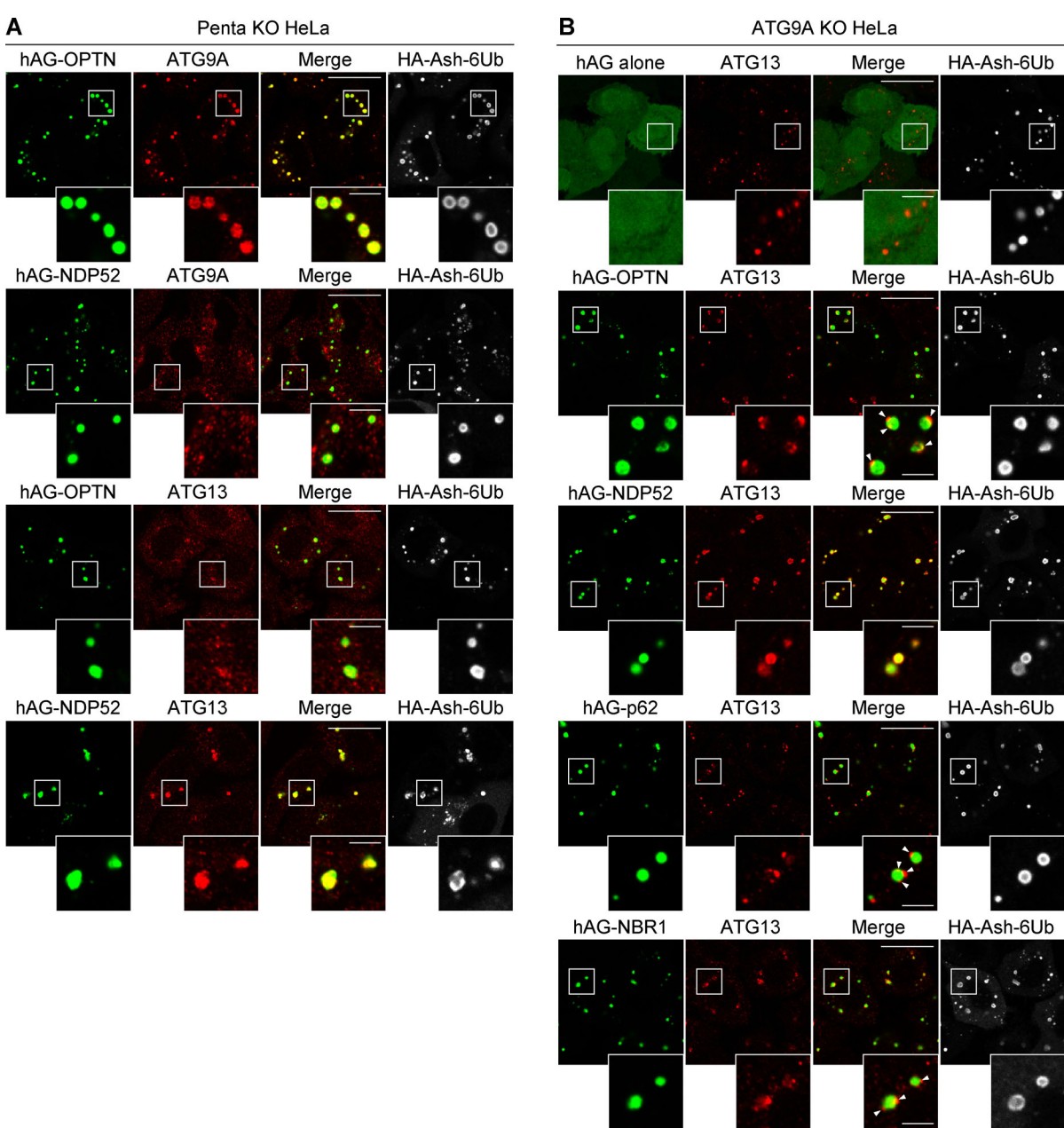

Figure S4. **OPTN and NDP52 Fluoppi foci contain ATG9A and ATG13, respectively. (A)** HA-Ash-6Ub– and hAG-tagged OPTN or NDP52 were transiently expressed in Penta KO HeLa cells. The cells were immunostained with anti-HA and anti-ATG9A or anti-ATG13 antibodies. Scale bars, 10 µm; insets, 2 µm. **(B)** HA-Ash-6Ub and hAG alone or hAG-tagged autophagy adaptors were transiently expressed in ATG9A KO HeLa cells. The cells were immunostained with anti-HA and anti-ATG13 antibodies. Scale bars, 10 µm; insets, 2 µm.

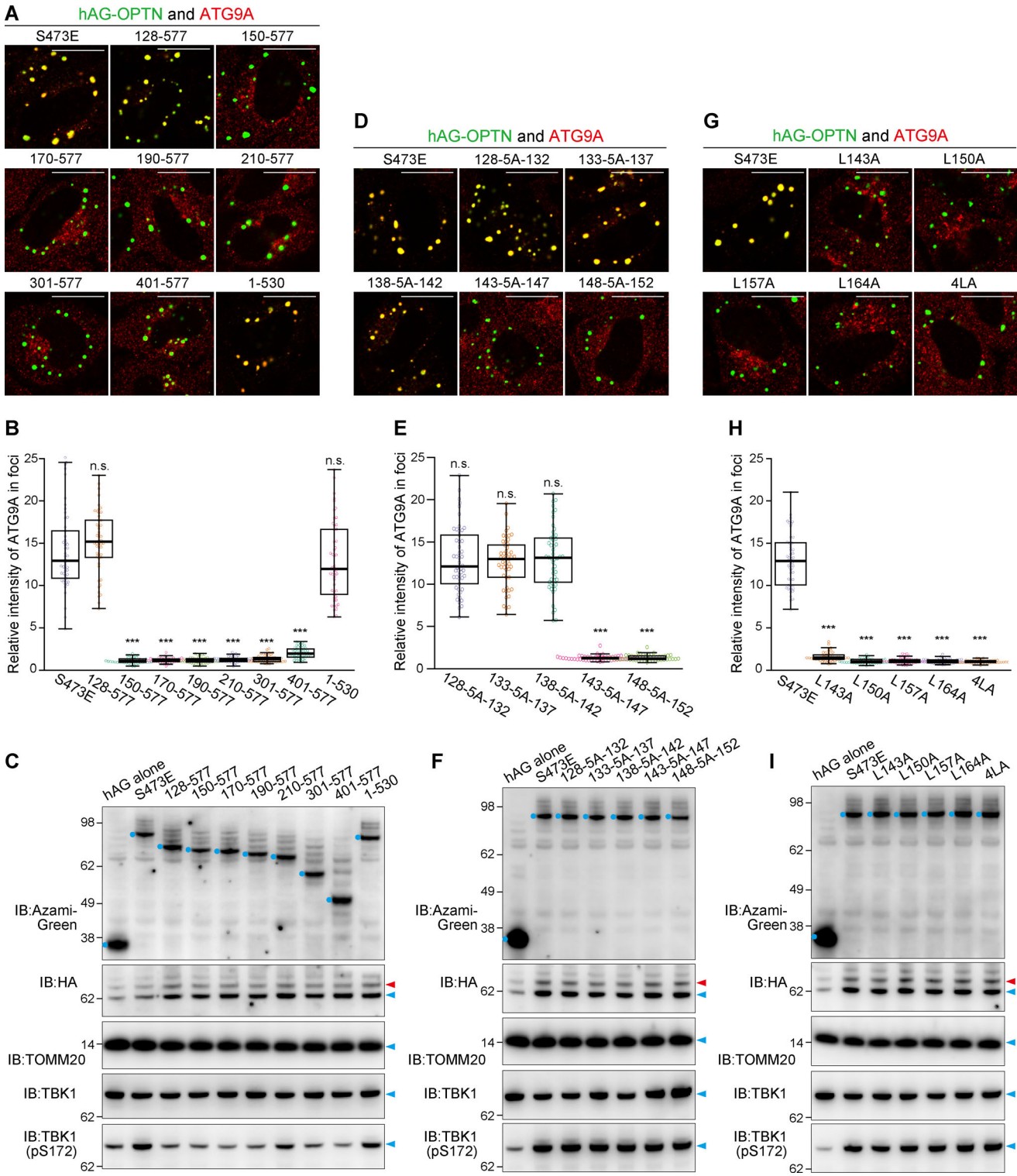

Figure S5. **Incorporation of ATG9A into the Fluoppi foci produced by OPTN mutants. (A, D, and G)** The indicated hAG-tagged OPTN mutants were transiently expressed in HeLa cells with HA-Ash-6Ub and then immunostained. The merged images of hAG-OPTN (green) and ATG9A immunostained (red) are shown. Scale bars, 10 μm. **(B, E, and H)** Efficiency of ATG9A incorporation into the mutant OPTN Fluoppi foci was quantified. Error bars represent mean ± SD, with 50 Fluoppi foci quantified in two independent experiments. Statistical differences were determined by a student's *t* test (n.s., not significant; ***, P < 0.001). **(C, F, and I)** Total cell lysates from HeLa cells transiently expressing the indicated hAG-tagged OPTN mutants and HA-Ash-6Ub were immunoblotted. hAG-OPTN and ubiquitinated HA-Ash-6Ub are indicated by the blue dots and red arrowheads, respectively.

