## [Peer Review File · The Journal of Cell Biology]

Critical Role of Mitochondrial Ubiquitination and the OPTN-ATG9A Axis in Mitophagy

Koji Yamano, Reika Kikuchi, Waka Kojima, Ryota Hayashida, Fumika Koyano, Junko Kawawaki, Takuji Shoda, Yosuke Demizu, Mikihiro Naito, Keiji Tanaka, and Noriyuki Matsuda

Corresponding Author(s): Koji Yamano, Tokyo Metropolitan Institute of Medical Science and Noriyuki Matsuda, Ubiquitin Project

Review Timeline:

Submission Date:	2020-01-05
Editorial Decision:	2020-02-07
Revision Received:	2020-04-02
Editorial Decision:	2020-04-23
Revision Received:	2020-05-06

Monitoring Editor: Li Yu

Scientific Editor: Marie Anne O'Donnell

Transaction Report:

DOI: <https://doi.org/10.1083/jcb.201912144>

February 7, 2020

Re: JCB manuscript #201912144

Dr. Koji Yamano
Tokyo Metropolitan Institute of Medical Science
Ubiquitin project
2-1-6 Kamikitazawa
Setagaya-ku, Tokyo 156-8506
Japan

Dear Dr. Yamano,

Thank you for submitting your manuscript entitled "Essential Roles of Ubiquitin Signals and the OPTN-ATG9A Axis in Mitochondria-Selective Autophagy". The manuscript was assessed by expert reviewers, whose comments are appended to this letter. We invite you to submit a revision if you can address the reviewers' key concerns, as outlined here.

You will see that the reviewers are very positive about the study and the new insight provided into the regulation of mitophagy but also provide some points to address to solidify the model proposed. In particular, a revision should focus on providing more insight into the interaction between OPTN and ATG9 versus TBK1, and discussion of the potential role of non-canonical autophagy adaptors in the mitophagy.

GENERAL GUIDELINES:

Text limits: Character count for an Article is < 40,000, not including spaces. Count includes title page, abstract, introduction, results, discussion, acknowledgments, and figure legends. Count does not include materials and methods, references, tables, or supplemental legends.

Figures: Articles may have up to 10 main text figures. Figures must be prepared according to the policies outlined in our Instructions to Authors, under Data Presentation, <http://jcb.rupress.org/site/misc/fora.xhtml>. All figures in accepted manuscripts will be screened prior to publication.

Supplemental information: There are strict limits on the allowable amount of supplemental data. Articles may have up to 5 supplemental figures. Up to 10 supplemental videos or flash animations are allowed. A summary of all supplemental material should appear at the end of the Materials and

methods section.

The typical timeframe for revisions is three months; if submitted within this timeframe, novelty will not be reassessed at the final decision. Please note that papers are generally considered through only one revision cycle, so any revised manuscript will likely be either accepted or rejected.

Thank you for this interesting contribution to Journal of Cell Biology. You can contact us at the journal office with any questions, cellbio@rockefeller.edu or call (212) 327-8588.

Sincerely,

Li Yu, Ph.D.
Monitoring Editor

Marie Anne O'Donnell, Ph.D.
Scientific Editor

Journal of Cell Biology

Reviewer #1 (Comments to the Authors (Required)):

In this manuscript Yamano and colleagues elaborate a tool to induce PINK/Parkin independent mitophagy through artificially ubiquitylated mitochondria. They nicely show that the PINK/Parkin system is essential for ubiquitination of damaged mitochondria, but is not required for autophagy activation. In addition, Yamano et al discover that OPTN binds to ATG9A vesicles during mitophagy and that this interaction is mutually exclusive with TBK1 binding.

The data in this manuscript is very well presented in the figures and clearly explained in the manuscript text. The new findings on how OPTN regulates mitophagy are interesting to a broad readership and merit publication in this journal. Prior to publication I would suggest the following improvements to the manuscript:

Major comments:

1. Why is there more TOMM20 and MTCO2 degradation with 2-Ub than with 4-Ub or 6-Ub? (Fig 1C, 1D). This is contrary to the introductory statements that Ub acts as a signal to recruit more PINK/Parkin which then would create even more ubiquitylation on the mitochondria.
2. The authors should test if the system of overexpressed OMM-Ub has similar mitophagy rates in cells where the protein concentrations of both PINK/Parkin can be regulated e.g. U2OS cells. Now the authors claim that OMM-Ub mitophagy is PINK independent, however they use cells that also lack Parkin (Fig. 1G).
3. When using OMM-UB WT K48R or K0, the authors should test the recruitment of autophagy adaptors (Fig 3D, E).
4. Can the authors speculate about their finding that LC3C is found much less in fluoppi foci of

NDP52, since NDP52 had been described as the major receptor for LC3C. In addition, it seems that LC3C is the main ATG8 being recruited to fluoppi foci.

5. Most crucial: Can the authors further strengthen their data that OPTN binding to TBK1 and ATG9A is mutually exclusive? It would be nice to further prove this by IPs from cell lysates, a Proximity ligation assay by IF or a competition assay with purified proteins. The authors should also induce mitophagy and then compare the binding of OPTN to ATG9A and TBK1. In addition, the authors should measure mitophagy induction (by mKeima FACS) with reconstituted OPTN but in the presence of ATG9A or TBK1 siRNAs.

Minor comments:

1. The authors should explain in the results section why only K0 ubiquitin overexpression induces mitophagy but not K48R ubiquitin (Fig. 1D) (instead of just mentioning it in the discussion).
2. Why are there increased levels of MTCO2 in T20-CII-3HA + SNIPER samples in comparison to - SNIPER (Fig. 2E)?
3. Direct interactions of the phosphorylated ubiquitin with cytosolic E3 ubiquitin ligase Parkin (Kazlauskaitė et al., 2015, Wauer et al., 2015, Yamano et al., 2015) recruits additional Parkin molecules to damaged mitochondria (Narendra et al., 2008), where PINK1 phosphorylates S65 in the ubiquitin-like domain of Parkin to activate Parkin E3 ligase activity (Kondapalli et al., 2012, Shiba-Fukushima et al., 2012).
4. Importantly, these Keima shifts were neutralized by bafilomycin A1, an inhibitor of lysosome function (Fig. 2 G and H).
5. Furthermore, the K63-linked ubiquitin was degraded through proteasomes by K48-linked branched chains (Ohtake et al., 2018).
6. HeLa, HEK293T, and HT1080 cell lines were cultured in DMEM supplemented with 10%(v/v) FBS,.....

Reviewer #2 (Comments to the Authors (Required)):

In the manuscript 'Essential Roles of Ubiquitin Signals and the OPTN-ATG9A Axis in Mitochondria-Selective Autophagy', Yamano et al. demonstrated that a known autophagy receptor OPTINEURIN (OPTN) targets damaged mitochondria via autophagy (mitophagy) by interacting with ATG9A. The interaction was confirmed in cells by using the fluoppi-tool showing that OPTN foci and ATG9A vesicles colocalize, which is abolished by mutating N-terminal leucine zipper (4LA mutant). Most importantly, they demonstrated that rescue of mitophagy-deficient Penta KO cells by OPTN 4LA mutant was largely suppressed indicating the interaction between OPTN and ATG9A plays an important role in mitophagy induction.

In addition, the authors initially aimed to address a very important question, whether PINK1 or PINK1-induced phosphor Ubiquitin activates autophagy machinery by engineering outer mitochondrial membrane (OMM)-Ubi in different form, showing that OMM- Ubi (K0) 2 expression is sufficient to induce mitophagy in cells. The key was to use KO (Lys null) mutant of ubiquitin which prevents to form branched chains. The event was independent of PINK1 determined by using PINK1 KO cells. In line with this, they also introduced a concept of SNIPER to induce ectopic ubiquitination by cIAP on mitochondria, which induced mitophagy. Since PINK1 Knockdown did not affect the effect supporting that the PARKIN-PINK1 system is dispensable for mitophagy in these conditions. The authors also observed something interesting: in cells, known autophagy receptors, OPTN and NDP52 forms a complex less efficiently than p62 and NBR1 although OPTN/NDP52 were more concentrated on mitochondria in close proximity to LC3B. These careful examinations led to the main question of this study, how OPTN contributes to mitophagy regulation. The study is very well constructed, and the manuscript is well written. The authors provided

sufficient and convincing evidence addressing an important question (although a part of their data may go against some of the previous studies): they showed a new aspect of regulatory mechanisms of mitophagy by OPTN ATG9A interaction.

I would only raise some minor points.

Minor points:

some wordings are not precise enough. For example,

1. abstract: directly interact with. Please change to OPTN forms a complex with ATG9A vesicles, or similar, since no direct biochemical data provided to show 'direct' interaction.
2. in general: it seems that the usage of the word 'essential' is not always proper. Please change to 'critical' or similar, since essential means that without it, the event cannot be occurred.
3. typing error in page 8, line 1. These Keima shifts>> shifts. Please correct.

Reviewer #3 (Comments to the Authors (Required)):

Yamano et al use innovative approaches to uncover the molecular role of OPTN during mitophagy. The authors show that ubiquitin is sufficient for mitophagy in the absence of PINK1 and Parkin highlighting that the fundamental role of PINK1 and Parkin in mitophagy is the generation of ubiquitin chains. The biggest conceptual advance of the paper is the discovery that OPTN directly interacts with Atg9a but not the Ulk1 complex. This is very interesting because it shows that NDP52 and OPTN can function differently to recruit the upstream autophagy machinery. The fluoppi system used to make the discovery is very innovative and could prove to be a useful tool for others to utilise in the autophagy field. Overall this is an excellent manuscript, with clear and convincing data, that makes an important contribution to the field. I think this work will be of broad interest to the readership of J Cell Biol. I have some comments/suggestions below which I think will help strengthen the manuscript and help to expand a little bit more on the author's discoveries.

1. Does the ubiquitin system used by the authors have a different dependence on autophagy receptors? For example, OPTN and NDP52 are the main receptors for PINK1 Parkin mitophagy - but under this Ub system can p62 drive mitophagy? It would be interesting to test whether p62 or NBR1 have any mitophagic activity using Penta KO cells expressing OMM-2Ub KO and rescued with an autophagy adaptor.

2. What does the translocation efficiency of OPTN look like in the presence of the different length Ubs and also the SNIPER system? Does the lower level of mitophagy observed with SNIPER and with longer Ub chains correlate with lower levels of OPTN or NDP52 recruitment? If so, this would suggest that OPTN or NDP52 have a preference for di-Ub on mitochondria rather than poly-Ub chains.

3. Figure 4D-G: I am not sure this figure adds much to the conclusions of the manuscript and could be moved to the supplement. Previous studies have shown that OPTN and NDP52 interact with ATG8s via co-IP and via binding experiments using recombinant proteins. Therefore, the main conclusion from this figure is that using the fluoppi experimental system interactions of ATG8s with OPTN and NDP52 are not readily detected. The discussion section should also reflect this point. Both Padman et al (2019) Nat Commun and Vargas et al (2019) Mol Cell have previously concluded that the LIR motif is not essential for mitophagy by the adaptors.

4. Can the authors show OPTN-Atg9a and OPTN 4LA binding via co-immunoprecipitation (with or

without mitophagy activation)? This experiment is not essential but would provide an additional line of evidence showing that OPTN and Atg9a directly interact. If possible, endogenous interaction would be preferable.

5. Is TBK1 activity required for OPTN-Atg9a binding? This can be tested using the fluoppi assay in the presence of the TBK1 inhibitor BX795. In addition, do disease associated mutations in OPTN affect OPTN-Atg9a binding?

6. It would be beneficial to show that Atg9a recruitment is indeed abolished in cells expressing OPTN 4LA during PINK1/Parkin mitophagy. This would help strengthen the author's conclusions.

Minor

1. Regarding the abstract, there are a number of manuscripts now that argue that autophagy adaptors function by promoting the recruitment the Ulk1 complex during mitophagy (e.g Lazarou et al (2015) Nature, Vargas et al (2019) Mol Cell) therefore the statement in the abstract that OPTN and NDP52 are thought to function mainly by binding to ATG8s is no longer current and should not be included. Indeed, a recent review by Mizushima supports this view (Mizushima (2020) Curr Opin Cell Biol).

2. Regarding the statement on page 5- "It remains a matter of debate whether PINK1 itself or PINK1-generated phosphorylated ubiquitin activates autophagy machinery". In the absence of Parkin, kinase dead PINK1 cannot recruit autophagy receptors and therefore cannot drive mitophagy pointing toward a role for s65 ub under conditions that have endogenous ubiquitin on mitochondria. However, in the context of the current study, it could be interpreted that high concentrations of ubiquitin on mitochondria overcome the requirement for PINK1.

3. It would be beneficial to note in the introduction that the function of OPTN and NDP52 in PINK1/Parkin mitophagy was initially shown to involve recruitment of upstream autophagy machinery, including the Ulk1 complex, by Lazarou et al (2015) Nature, which was then followed up in Vargas et al (2019) Mol Cell showing that NDP52 can directly interact with the FIP200 subunit of the Ulk1 complex.

**TOKYO METROPOLITAN INSTITUTE
OF MEDICAL SCIENCE**

Koji Yamano
Ubiquitin Project
2-1-6 Kamikitazawa
Setagaya-ku, Tokyo 156-8506
JAPAN
Phone: +81-3-5316-3123
Fax: +81-3-5316-3152
E-mail:yamano-kj@igakuken.or.jp

April 2, 2020

Revised version for JCB manuscript #201912144 "Essential Roles of Ubiquitin Signals and the OPTN-ATG9A Axis in Mitochondria-Selective Autophagy" (the original title)"

Professor Li Yu,
JCB Monitoring Editor

Dear Yu,

We would like to thank you and the reviewers for the comments and suggestions, many of which improved our manuscript. We have performed a number of the suggested experiments, added new data and modified the text to address the critiques. In particular, we focused our revisions on the relationship between TBK1 and ATG9A on OPTN binding, and the role of autophagy adaptors following OMM-Ub-induced mitophagy. Our point-by-point responses to the reviewers' comments follow.

Reviewer #1 (Comments to the Authors (Required)):

In this manuscript Yamano and colleagues elaborate a tool to induce PINK/Parkin independent mitophagy through artificially ubiquitylated mitochondria. They nicely show that the PINK/Parkin system is essential for ubiquitination of damaged mitochondria, but is not required for autophagy activation. In addition, Yamano et al discover that OPTN binds to ATG9A vesicles during mitophagy and that this interaction is mutually exclusive with TBK1 binding.

The data in this manuscript is very well presented in the figures and clearly explained in the manuscript text. The new findings on how OPTN regulates mitophagy are interesting to a broad readership and merit publication in this journal. Prior to publication I would suggest the following improvements to the manuscript:

Major comments:

1. Why is there more TOMM20 and MTCO2 degradation with 2-Ub than with 4-Ub or 6-Ub? (Fig 1C, 1D). This is contrary to the introductory statements that Ub acts as a signal to recruit more PINK/Parkin which then would create even more ubiquitylation on the mitochondria.

As shown in Figure 1 B, the expression level of 2-Ub K0 is much higher than that of 4-Ub K0 and 6-Ub K0. Furthermore, our revised experiment (please see responses to reviewer #1 comment No.3, and reviewer #3 comments No.1 and 2) showed that OPTN is the adaptor responsible for mitochondrial degradation in response to OMM-Ub mitophagy (new Figure 1 G and H). Previous structural analysis

(Nakazawa et al. 2016 Nat Commun) showed that two-tandem ubiquitin is sufficient for recognition by the UBAN domain of OPTN. From these results, we think that the amount of linear chain on the mitochondria rather than the length of each linear chain is critical for OPTN recruitment and subsequent mitochondrial degradation in response to OMM-Ub-induced mitophagy.

We agree with the reviewer that Parkin promotes more ubiquitination through the positive feedback amplification loop. Previous studies showed that Parkin does not have substrate specificity (Sarraf et al. 2013 Nature and Koyano et al. 2019 JBC), which means that activated Parkin increases the amount of ubiquitin rather than elongates the length of ubiquitin on a particular substrate. Therefore, a critical role for increased ubiquitin signals on mitochondria is common to both OMM-Ub-induced mitophagy and Parkin-mediated mitophagy.

2. The authors should test if the system of overexpressed OMM-Ub has similar mitophagy rates in cells where the protein concentrations of both PINK/Parkin can be regulated e.g. U2OS cells. Now the authors claim that OMM-Ub mitophagy is PINK independent, however they use cells that also lack Parkin (Fig. 1G).

We thank the reviewer for this constructive comment. We previously reported that the HEK293 cell line expresses endogenous Parkin and that the mitochondrial protein HK1 is ubiquitinated by endogenous Parkin upon dissipation of the mitochondrial membrane potential (Okatsu et al. 2012 BBRC). In the revised experiment, we thus compared OMM-Ub-induced mitophagy and Parkin/PINK1-mediated mitophagy using two different cell lines, HEK293 and HeLa cells. Mitophagy was again measured using a Keima-FACS assay. Please see “Figure 1 for reviewers” below.

In HEK293 with endogenous Parkin, mitophagy induced by AO for 6hrs was very low (9.1% compared to 7.6% in the DMSO control). In the presence of overexpressed YFP-Parkin, the mitophagy rate after 6hrs with AO was ~ 24%, indicating that, even though HEK293 expresses endogenous Parkin, the Parkin-mediated mitophagy rate is relatively low. In sharp contrast, the mitophagy rate in HeLa cells overexpressing YFP-Parkin after 6hrs with AO was very high (96.4%), whereas mitophagy was not observed in the absence of exogenous YFP-Parkin expression (4.7% with AO for 6hrs). Under these conditions, OMM-Ub-induced mitophagy was 25.5% with DMSO and 25.0% after 6hrs with AO in HEK293 cells and was 61.1% with DMSO and 72.7% after 6hrs with AO in HeLa cells. These results indicate: 1) mitophagy rates in cells with overexpressed OMM-Ub are comparable to those induced by the Parkin/PINK1 system (25.0% vs 24.0% in HEK293 and 72.7% vs 96.4% in HeLa), and 2) endogenous Parkin does not seem to contribute to OMM-Ub-induced mitophagy. Wider interpretations of these results are complicated by potential differences in the cell lines including transfection efficiencies of the OMM-Ub plasmids, the levels of PINK1 accumulation on damaged mitochondria, counteraction activity by de-Ubase, and basal autophagy activation. We, therefore, prefer to limit these results to this letter for the reviewers.

As mentioned in the main text, and pointed out by the reviewer, the HeLa cell line genetically lacks Parkin. Using HeLa cells, our original results (Figure 1 D and E) showed that OMM-Ub-induced mitophagy occurs in a Parkin-independent manner. A previous report (Lazarou et al. 2015 Nature) indicated that PINK1 can activate mitophagy independently of Parkin. To further explore this, we tested if OMM-Ub-induced mitophagy is PINK1 independent or not, and found that OMM-Ub-induced mitophagy occurs in PINK1 KO cells (Figure 1 I). We would like to emphasize that Parkin always needs PINK1 for its E3 activation. Therefore without PINK1, Parkin is unable to activate. This PINK1-Parkin cascade (PINK1 functions upstream of Parkin) has been shown in vitro and in vivo (Clark et al. 2006

Nature, Park et al. 2006 Nature, Matsuda et al. 2010 JCB, Narendra et al. 2010 PLoS Biol, Lazarou et al. 2012 Dev Cell). Thus, we concluded that neither Parkin nor PINK1 is required for OMM-Ub-induced mitophagy. To further clarify the PINK1-Parkin cascade for readers, we explicitly state in the Introduction that “PINK1 acts upstream of Parkin”.

Figure 1 for reviewers

(A and B) YFP, YFP-Parkin, or OMM-2Ub K0 were expressed in HEK293 cells (A) and HeLa cells (B) with mt-Keima. The cells were then treated with or without antimycin A/oligomycin (AO) for 6hrs, and subjected to Keima-FACS assay. Representative FACS data with the percentage of cells exhibiting lysosomal positive mt-Keima are shown.

(C and D) Quantification of the FACS-based mitophagy in A and B. Error bars represent mean \pm s.d. of three independent experiments.

3. When using OMM-UB WT K48R or K0, the authors should test the recruitment of autophagy adaptors (Fig 3D, E).

We thank the reviewer for the comment. According to the reviewer's suggestion, we expressed OMM-2Ub WT and OMM-2Ub K0 constructs in Penta KO HeLa cells stably expressing 3FLAG-tagged versions of each autophagy adaptor (OPTN, NDP52, p62 and NBR1). Interestingly, when OMM-2Ub WT was expressed, no mitochondrial recruitment was observed for any of the autophagy adaptors (new Figure 1 G). In contrast, OMM-2Ub K0 expression did induce the mitochondrial recruitment of OPTN, p62, and NBR1, but not NDP52 (new Figure 1 G). In addition, based on suggestions from reviewer #3, we used the FACS-based assay to also determine which autophagy adaptors recovered OMM-Ub-induced mitophagy in Penta KO cells. Of the adaptors tested, only OPTN recovered OMM-Ub-induced mitophagy (new Figure 1 H and new Supplemental Figure 1 C). The reason why NDP52 was not recruited to mitochondria by the expression of OMM-2Ub K0 is currently unclear, but we speculate that the NDP52 ubiquitin binding domain may have a lower affinity for the linear Ub-chains. We reflect on these results in the main text and have incorporated them into both Figure 1 and Supplemental Figure 1. Thanks to the reviewer's suggestion, we believe that our understanding of the role OPTN plays in OMM-Ub-induced mitophagy has been strengthened.

4. Can the authors speculate about their finding that LC3C is found much less in fluoppi foci of NDP52, since NDP52 had been described as the major receptor for LC3C. In addition, it seems that LC3C is the main ATG8 being recruited to fluoppi foci.

As indicated by the reviewer, a previous study (Muhlinen et al 2012 Mol Cell) used co-immunoprecipitation and luciferase-based assays to show that NDP52 was selectively bound to LC3C. We think that this selective NDP52-LC3C interaction among the ATG8 homologues was also observed in our fluoppi assay. Please see "Figure 2 for reviewers", which is essentially the same as Figure 4 G, but focuses on the NDP52-ATG8 interactions. Although the binding affinities of NDP52 for ATG8s are relatively low, the NDP52-LC3C interaction is the highest among the ATG8 family proteins.

Figure 2 for reviewers

Efficiency of ATG8s incorporation into the NDP52 fluoppi foci.
Original data are derived from Figure 4G.

Based on our fluoppi data, the affinity of NDP52 for ATG8s was relatively weak when compared to other autophagy adaptors such as p62. Although the exact reason is currently unclear, we speculate that it reflects differences in the assay systems. Our fluoppi assay is cell-based and occurs in a ubiquitin-dependent manner. For example, the OPTN-TBK1 axis is activated in the presence of ubiquitin chains, which enhance ATG8 binding via phosphorylation of the LIR in OPTN (Heo et al. Mol Cell 2015). p62 has also been reported to undergo phosphorylation when it forms liquid droplets through oligomerization with ubiquitin chains (Matsumoto et al. Mol Cell 2011, Ichimura et al. Mol Cell 2013). As a consequence, the binding affinities of OPTN-ATG8s and p62-ATG8s in our fluoppi assay

will be different from those previously reported using ubiquitin-independent immunoprecipitation. To exclude any contribution by other autophagy machinery, such as NDP52-FIP200 (Ravenhill et al. 2019 Mol Cell, and Vergas et al. 2019, Mol Cell) and OPTN-ATG9A (this study), we used an ATG5 KO HeLa cell line. We speculate that one or more of these reasons contributed to the differences in the binding affinity of ATG8s relative to previous studies.

5. Most crucial: Can the authors further strengthen their data that OPTN binding to TBK1 and ATG9A is mutually exclusive? It would be nice to further prove this by IPs from cell lysates, a Proximity ligation assay by IF or a competition assay with purified proteins. The authors should also induce mitophagy and then compare the binding of OPTN to ATG9A and TBK1. In addition, the authors should measure mitophagy induction (by mKeima FACS) with reconstituted OPTN but in the presence of ATG9A or TBK1 siRNAs.

We thank the reviewer for the constructive comment. Accordingly, we performed co-IP experiments and tried to compare OPTN binding to TBK1 and ATG9A by IP (new Figure 6 G and “Figure 3 for reviewers”). We initially tried to pull down OPTN fluoppi foci consisting of 3FLAG-hAG-OPTN and HA-Ash-6Ub with an anti-FLAG antibody after the cells were solubilized with detergent (1% digitonin or 1% ND-40), but neither TBK1 nor ATG9A were co-immunoprecipitated with OPTN (data not shown). These results combined with fluorescent microscopy imaging revealed that the OPTN fluoppi foci dissociated soon after solubilization. We next tried to fix OPTN fluoppi foci using a DSP crosslinker, but because of random chemical crosslinking, we were unable to efficiently extract the fluoppi foci with 1% digitonin (data not shown). Finally, using DSP-crosslinking, 1% NP-40 solubilization, followed by sonication and immunoprecipitation with an anti-FLAG tag gel, we were able to pull down OPTN fluoppi foci with TBK1 and ATG9A. Please see new Figure 6 G. Using this method, we have shown that the OPTN 4LA mutant is unable to pull down ATG9A, but can pull down TBK1 with a similar efficiency to that of OPTN WT. Conversely, the OPTN M44Q/L54Q mutant cannot pull down TBK1, but can pull down ATG9A with a similar efficiency to that of OPTN WT.

Figure 3 for reviewers

(A) HeLa cells stably expressing Parkin and 3FLAG-OPTN were treated with 100 nM Bafilomycin A1 in the presence or absence of valinomycin for 2hrs. The cells were solubilized with 1% digitonin and incubated with anti-FLAG antibody-conjugated gel. The gel was washed three times, and the bound proteins were eluted with SDS-PAGE sample buffer. The bound, 5% of input and 5% of flow through (F.T.) fractions were analyzed by immunoblotting.

(B) The cells in (A) were treated with 1 mM DSP for 30min at room temperature. After quenching the reaction with 200 mM Tris-containing buffer, the cells were solubilized with 1% digitonin and incubated with anti-FLAG antibody-conjugated gel. Proteins were analyzed by immunoblotting. Red arrowheads denote oligomeric protein bands possibly by the crosslinking.

Furthermore, based on the suggestion from reviewer #3, we also show that ATG9A incorporation into OPTN foci is not enhanced or blocked by a TBK1 inhibitor (new Figure 6 C-F), or by *TBK1* gene deletion (new Figure 6 H-J). These results indicate that neither TBK1 activation nor TBK1 itself affect the OPTN-ATG9A interaction, and that the interactions are likewise not affected by the OPTN mutant being unable to bind ATG9A. Although we also sought to use IP to examine the OPTN-ATG9A interaction during Parkin-mediated mitophagy, the likely low binding affinity prevented further exploration of this interaction (Please see “Figure 3 for reviewers” and our response to reviewer #3 comment No.4). These findings demonstrate a clear relationship in OPTN binding between TBK1 and ATG9A. These new data have been added to the results section of the main text and Figure 6.

Minor comments:

1. The authors should explain in the results section why only K0 ubiquitin overexpression induces mitophagy but not K48R ubiquitin (Fig. 1D) (instead of just mentioning it in the discussion).

We have added the text “... inhibition of branched chain formation is an important trigger for mitophagy” to the results section as suggested.

2. Why are there increased levels of MTCO2 in T20-CII-3HA + SNIPER samples in comparison to - SNIPER (Fig. 2E)?

We carefully re-performed the experiments (N=3) in Fig. 2E to see whether the levels of MTCO2 actually increased with SNIPER treatment in cells expressing T20-CII-3HA. As shown in “Figure 4 for reviewers” below, the MTCO2 levels did not change in response to SNIPER treatment. To avoid any ambiguity, we have replaced the original data (Fig. 2E, T20-CII-3HA section) with the new results.

Figure 4 for Reviewers

HT1080 cells stably expressing T20-CII-3HA were treated with or without SNIPER(CRABP)-11 for 9hrs. Total cell lysates were analyzed by immunoblotting.

3. Direct interactions of the phosphorylated ubiquitin with cytosolic E3 ubiquitin ligase Parkin (Kazlauskaite et al., 2015, Wauer et al., 2015, Yamano et al., 2015) recruits additional Parkin molecules to damaged mitochondria (Narendra et al., 2008), where PINK1 phosphorylates S65 in the ubiquitin-like domain of Parkin to activate Parkin E3 ligase activity (Kondapalli et al., 2012, Shiba-Fukushima et al., 2012).

We thank the reviewer for the comment. We have changed the original sentence to “Direct interactions between the phosphorylated ubiquitin and the E3 ubiquitin ligase Parkin recruit cytosolic Parkin to damaged mitochondria”.

4. Importantly, these Keima shifts were neutralized by bafilomycin A1, an inhibitor of lysosome function (Fig. 2 G and H).

Changed accordingly.

5. Furthermore, the K63-linked ubiquitin was degraded through proteasomes by K48-linked branched chains (Ohtake et al., 2018).

Changed accordingly.

6. HeLa, HEK293T, and HT1080 cell lines were cultured in DMEM supplemented with 10%(v/v) FBS,.....

Changed accordingly.

Reviewer #2 (Comments to the Authors (Required)):

In the manuscript 'Essential Roles of Ubiquitin Signals and the OPTN-ATG9A Axis in Mitochondria-Selective Autophagy', Yamano et al. demonstrated that a known autophagy receptor OPTINEURIN (OPTN) targets damaged mitochondria via autophagy (mitophagy) by interacting with ATG9A. The interaction was confirmed in cells by using the fluoppi-tool showing that OPTN foci and ATG9A vesicles colocalize, which is abolished by mutating N-terminal leucine zipper (4LA mutant). Most importantly, they demonstrated that rescue of mitophagy-deficient Penta KO cells by OPTN 4LA mutant was largely suppressed indicating the interaction between OPTN and ATG9A plays an important role in mitophagy induction.

In addition, the authors initially aimed to address a very important question, whether PINK1 or PINK1-induced phosphor Ubiquitin activates autophagy machinery by engineering outer mitochondrial membrane (OMM)-Ubi in different form, showing that OMM- Ubi (K0) 2 expression is sufficient to induce mitophagy in cells. The key was to use KO (Lys null) mutant of ubiquitin which prevents to form branched chains. The event was independent of PINK1 determined by using PINK1 KO cells. In line with this, they also introduced a concept of SNIPER to induce ectopic ubiquitination by cIAP on mitochondria, which induced mitophagy. Since PINK1 Knockdown did not affect the effect supporting that the PARKIN-PINK1 system is dispensable for mitophagy in these conditions. The authors also observed something interesting: in cells, known autophagy receptors, OPTN and NDP52 forms a complex less efficiently than p62 and NBR1 although OPTN/NDP52 were more concentrated on mitochondria in close proximity to LC3B. These careful examinations led to the main question of this study, how OPTN contributes to mitophagy regulation.

The study is very well constructed, and the manuscript is well written. The authors provided sufficient and convincing evidence addressing an important question (although a part of their data may go against some of the previous studies): they showed a new aspect of regulatory mechanisms of mitophagy by OPTN ATG9A interaction.

We thank the reviewer for their positive evaluation of our manuscript. We have corrected the issues raised by the reviewer accordingly.

I would only raise some minor points.

Minor points:

some wordings are not precise enough. For example,

1. abstract: directly interact with. Please change to OPTN forms a complex with ATG9A vesicles, or similar, since no direct biochemical data provided to show 'direct' interaction.

We changed the expression “directly interacts with” to “forms a complex with” in the abstract. We also changed similar terminology in other sections of the manuscript.

2. in general: it seems that the usage of the word 'essential' is not always proper. Please change to 'critical' or similar, since essential means that without it, the event cannot be occurred.

We have corrected the text accordingly. We also changed the manuscript title to “Critical Roles of Ubiquitin Signals and the OPTN-ATG9A Axis in Mitochondria-Selective Autophagy”.

3. typing error in page 8, line 1. These Keima shifts>> shifts. Please correct.

Changed accordingly.

Reviewer #3 (Comments to the Authors (Required)):

Yamano et al use innovative approaches to uncover the molecular role of OPTN during mitophagy. The authors show that ubiquitin is sufficient for mitophagy in the absence of PINK1 and Parkin highlighting that the fundamental role of PINK1 and Parkin in mitophagy is the generation of ubiquitin chains. The biggest conceptual advance of the paper is the discovery that OPTN directly interacts with Atg9a but not the Ulk1 complex. This is very interesting because it shows that NDP52 and OPTN can function differently to recruit the upstream autophagy machinery. The fluoppi system used to make the discovery is very innovative and could prove to be a useful tool for others to utilise in the autophagy field. Overall this is an excellent manuscript, with clear and convincing data, that makes an important contribution to the field. I think this work will be of broad interest to the readership of J Cell Biol. I have some comments/suggestions below which I think will help strengthen the manuscript and help to expand a little bit more on the author's discoveries.

1. Does the ubiquitin system used by the authors have a different dependence on autophagy receptors? For example, OPTN and NDP52 are the main receptors for PINK1 Parkin mitophagy - but under this Ub system can p62 drive mitophagy? It would be interesting to test whether p62 or NBR1 have any mitophagic activity using Penta KO cells expressing OMM-2Ub KO and rescued with an autophagy adaptor.

We thank the reviewer for the constructive comment. Based on this suggestion (and a suggestion from reviewer #1), we expressed OMM-2Ub WT and OMM-2Ub K0 in Penta KO HeLa cells stably expressing the autophagy adaptors (OPTN, NDP52, p62 and NBR1) with a 3FLAG-tag. When OMM-2Ub WT was expressed, no mitochondrial recruitment was observed for any of the autophagy adaptors (new Figure 1 G). In contrast, OMM-2Ub K0 expression induced mitochondrial recruitment of OPTN, p62, and NBR1, but not NDP52 (new Figure 1 G). We also used the FACS-based assay to determine which autophagy adaptors could recover OMM-Ub-induced mitophagy in Penta KO cells. Only OPTN was able to recover OMM-Ub-induced mitophagy (new Figure 1 H and new Supplemental Figure 1 C). While p62 and NBR1 were efficiently recruited to mitochondria by OMM-2Ub K0 expression, they could not recover mitophagy. We reflect on these results in the main text and have incorporated them into Figure 1 and Supplemental Figure 1. Thanks to the reviewer's suggestion, we believe that our understanding of the role OPTN plays in both Parkin-mediated mitophagy and OMM-Ub-induced mitophagy has been strengthened.

2. What does the translocation efficiency of OPTN look like in the presence of the different length Ubs and also the SNIPER system? Does the lower level of mitophagy observed with SNIPER and with longer Ub chains correlate with lower levels of OPTN or NDP52 recruitment? If so, this would suggest that OPTN or NDP52 have a preference for di-Ub on mitochondria rather than poly-Ub chains.

As indicated in our response to reviewer #1 comment No.1, the expression level of 2-Ub K0 is much higher than that of 4-Ub K0 and 6-Ub (Figure 1B), and we think that this is the main reason why 2-Ub has a higher mitophagy rate than 4-Ub or 6-Ub. Although NDP52 cannot be recruited to mitochondria containing 2-Ub K0 (new Figure 1 G) and cannot recover OMM-induced mitophagy in Penta KO cells (new Figure 1H), OPTN, which can bind di-Ub (Nakazawa et al. 2016 Nat Commun), was found to be responsible for mitochondrial clearance in response to OMM-Ub-induced mitophagy. For the SNIPER system, most of the CRABP-II proteins were degraded through the proteasomes (Okuhira et al. 2017 Mol Pharmacol), and no obvious ubiquitin signals accumulated on the mitochondria. We speculate that the lower ubiquitin signal on mitochondria accounts for the relatively low efficiency of SNIPER-induced mitophagy compared to OMM-Ub-induced mitophagy or Parkin-mediated mitophagy.

3. Figure 4D-G: I am not sure this figure adds much to the conclusions of the manuscript and could be moved to the supplement. Previous studies have shown that OPTN and NDP52 interact with ATG8s via co-IP and via binding experiments using recombinant proteins. Therefore, the main conclusion from this figure is that using the fluoppi experimental system interactions of ATG8s with OPTN and NDP52 are not readily detected. The discussion section should also reflect this point. Both Padman et al (2019) Nat Commun and Vargas et al (2019) Mol Cell have previously concluded that the LIR motif is not essential for mitophagy by the adaptors.

We would like to emphasize that our fluoppi assay for quantifying the interactions between autophagy adaptors and ATG8 family proteins is experimentally different from previous studies. Although the conclusions from Figure 4 D-G are not novel, the uniqueness of the fluoppi assay compared to previous experimental designs bears mentioning. Because the fluoppi assay is cell-based, ubiquitin-dependent autophagy adaptor-ATG8 interactions can be investigated. In addition, utilization

of the ATG5 KO cell line excluded any possible involvement from other autophagy core units such as NDP52-FIP200 and OPTN-ATG9A interactions. Furthermore, JCB does not allow for more than five supplemental figures. We are currently at that limit with supplemental figures consisting of raw data and/or supporting data related to the main figures. For these reasons, we would prefer that the figure remain in the main text. We reflect on the reviewer's suggestion and have cited the previous studies, Padman et al. 2019 Nat Commun and Vargas et al. 2019 Mol Cell, in the Discussion.

4. Can the authors show OPTN-Atg9a and OPTN 4LA binding via co-immunoprecipitation (with or without mitophagy activation)? This experiment is not essential but would provide an additional line of evidence showing that OPTN and Atg9a directly interact. If possible, endogenous interaction would be preferable.

We thank the reviewer for the suggestion. To determine if OPTN-ATG9A interactions during mitophagy could be demonstrated by co-immunoprecipitation, HeLa cells stably expressing Parkin and 3FLAG-OPTN were treated with valinomycin for 3hrs, solubilized with 1% digitonin, and proteins were co-immunoprecipitation with an anti-FLAG antibody. Please see "Figure 3 for reviewers" in our response to reviewer #1 comment No.5. As shown in 3A, FLAG-OPTN itself was efficiently immunoprecipitated (almost 100% of OPTN was in the bound fraction), but ATG9A was not present. Similarly, neither LC3B nor WIPI2 were co-immunoprecipitated. TBK1 was present in the bound fraction, but the amount was negligible (~ 1%). We next treated the cells with a chemical crosslinker (1mM DSP), as reported by Vargas et al. 2019 Mol Cell, and then repeated the co-immunoprecipitation. As shown in 3B, the immunoprecipitation efficiencies of TBK1 as well as the activated form of TBK1 were greatly improved. However, under this condition, the amount of ATG9A in the bound fraction remained limiting (< 1%) and no difference was observed in the presence or absence of valinomycin. Although the lipidated form of LC3B was specifically co-immunoprecipitated by valinomycin treatment, the amount was < 1%. Despite these efforts, we were unable to show mitophagy-dependent OPTN-ATG9 interaction by co-immunoprecipitation.

As an alternative to the mitophagy co-IP, we tried to pull down OPTN fluoppi foci consisting of 3FLAG-tagged hAG-OPTN and HA-Ash-6Ub with an anti-FLAG antibody. As shown in "response to reviewer#1 comment No.5", we successfully co-immunoprecipitated ATG9A with 3FLAG-hAG-OPTN in a leucine zipper-dependent manner. This result has been added into Figure 6 G. In addition to the fluoppi assay, we also provide evidence for leucine zipper-dependent OPTN-ATG9A interactions using the FRB-FKBP system. This result has been added as Figure 6 K-M.

5. Is TBK1 activity required for OPTN-Atg9a binding? This can be tested using the fluoppi assay in the presence of the TBK1 inhibitor BX795. In addition, do disease associated mutations in OPTN affect OPTN-Atg9a binding?

We thank the reviewer for the suggestion. To determine if TBK1 activity is required for OPTN-ATG9A interactions, we examined ATG9A incorporation into OPTN fluoppi foci after BX-795 treatment. Please see new Figure 6 C-F. While the activated TBK1 (pS172) signal was almost completely absent in the OPTN foci in cells treated with BX-795, the recruitment of ATG9A vesicles to the foci was comparable to cells treated with DMSO. In addition, when we used the hAG-OPTN M44Q/L54Q mutant, which is unable to interact with TBK1 (Li et al. 2016 Nat Commun), we found that TBK1 was excluded from the

OPTN foci and that this exclusion did not affect ATG9A recruitment. To further examine this, we generated a *TBK1*^{-/-} HCT116 cell line (new Figure 6 H) and found that ATG9A incorporation was not affected by *TBK1* gene depletion (new Figure 6 I and J). Taken together, these results indicate that *TBK1* activity is not required for OPTN-ATG9A binding.

As indicated by the reviewer, multiple disease-associated mutations have been reported for OPTN. To examine their potential effects on OPTN-ATG9A interactions, we constructed a series of hAG-OPTN plasmids harboring known mutations in glaucoma patients (H26D, E50K, E103D, T202R, A336G, A377T, and H486R), amyotrophic lateral sclerosis (ALS) patients (G159V, V161M, Q454E, and E478G), and an artificial mutant (I463I/L464A) expected to disrupt ubiquitin binding.

Figure 5 for reviewers

(A) Schematic diagram of the OPTN domain architecture and disease-associated mutations (glaucoma in red and ALS in green).

(B) HeLa cells expressing hAG-OPTN WT or indicated mutants and HA-Ash-6Ub were immunostained. Bars, 10 μ m.

(C) hAG-OPTN expression was confirmed by immunoblotting.

Please see Figure 5 for reviewers. Cells expressing glaucoma pathogenic mutants had OPTN fluoppi foci with ATG9A vesicles, whereas cells expressing some of the ALS mutants (Q454E and E478G) exhibited defects in fluoppi foci formation, likely due to disrupted ubiquitin binding (Li et al. 2017 Autophagy). Although the G159V and V161M disease mutations are located in the leucine zipper domain, the associated OPTN foci still contained ATG9A vesicles. Based on these results, we concluded that none of the known disease mutations in OPTN that we tested disrupt ATG9A interactions. Since these results were negative, we would prefer that they be made available only to the reviewers.

6. It would be beneficial to show that Atg9a recruitment is indeed abolished in cells expressing OPTN 4LA during PINK1/Parkin mitophagy. This would help strengthen the author's conclusions.

We concur with the reviewer's comment and consequently performed an additional experiment to demonstrate OPTN-ATG9 interactions during Parkin-mediated mitophagy. Penta KO cells stably expressing Parkin and 3FLAG-tagged OPTN (WT, 4LA, F178A, or 4LA/F178A) were treated with valinomycin for 3hrs and then immunostained with anti-FLAG, anti-ATG9A and anti-HSP60 antibodies. The recruitment of ATG9A to mitochondria was then quantified. Please see new Figure 7 E and F. We found that Parkin alone did not induce ATG9A mitochondrial recruitment. In contrast, the recruitment of ATG9A to mitochondria during mitophagy was recovered by OPTN WT (as well as the F178A mutant), and ATG9A recruitment was completely abolished by the OPTN 4LA mutants. Furthermore, although mitochondrial clearance in Penta KO cells can be recovered by NDP52 (Lazarou et al. 2015 Nature), it did not induce robust ATG9A recruitment. These results indicate that OPTN-ATG9A interactions occur during Parkin-mediated mitophagy and that the 4LA mutation in OPTN abrogates ATG9A recruitment to mitochondria during Parkin-mediated mitophagy.

Minor

1. Regarding the abstract, there are a number of manuscripts now that argue that autophagy adaptors function by promoting the recruitment the Ulk1 complex during mitophagy (e.g Lazarou et al (2015) Nature, Vargas et al (2019) Mol Cell) therefore the statement in the abstract that OPTN and NDP52 are thought to function mainly by binding to ATG8s is no longer current and should not be included. Indeed, a recent review by Mizushima supports this view (Mizushima (2020) Curr Opin Cell Biol).

ATG8 binding ability in OPTN is not essential for mitophagy, but rather has a supporting effect since introduction of the F178A mutation into OPTN 4LA completely blocked mitophagy (Figure 7). We prefer to keep the original sentence "in addition to binding ATG8 proteins, the crucial/critical autophagy adaptors possess additional binding sites for autophagy core units..." in the abstract, and have addressed the reviewer's comment in the Discussion.

2. Regarding the statement on page 5- 'It remains a matter of debate whether PINK1 itself or PINK1-generated phosphorylated ubiquitin activates autophagy machinery'. In the absence of Parkin, kinase dead PINK1 cannot recruit autophagy receptors and therefore cannot drive mitophagy pointing toward a role for s65 ub under conditions that have endogenous ubiquitin on mitochondria. However,

in the context of the current study, it could be interpreted that high concentrations of ubiquitin on mitochondria overcome the requirement for PINK1.

We thank the reviewer for the comment. As reported by Wauer et al. (2014, EMBO J), although phosphorylation of S65 in ubiquitin has inhibitory effects on E2/E3 systems and DUB cleavage, the phosphorylated ubiquitin can still be recognized by UBDs. Therefore, the phosphorylated ubiquitin generated by PINK1 overexpression might stably accumulate on the mitochondrial surface, which is then preferentially recognized by the autophagy adaptors. Although we did not test this possibility in the current study, it would be interesting for a future study.

3. It would be beneficial to note in the introduction that the function of OPTN and NDP52 in PINK1/Parkin mitophagy was initially shown to involve recruitment of upstream autophagy machinery, including the Ulk1 complex, by Lazarou et al (2015) Nature, which was then followed up in Vargas et al (2019) Mol Cell showing that NDP52 can directly interact with the FIP200 subunit of the Ulk1 complex.

The reviewer's comment has been addressed in the Introduction.

During preparation of the revised manuscript, we noticed that the bar graphs (OMM-2Ub K0 AO for 6hrs and OMM-6Ub K0 DMSO) in the original Fig 1G (now Fig 1I) were switched. This is now corrected. We apologize for the error and would like to emphasize that this does not affect the interpretation of the results.

Modifications and changes in the revised manuscript are highlighted in yellow. We hope that we have adequately addressed the reviewer comments and that our manuscript is now suitable for publication in Journal of Cell Biology.

Sincerely,

Koji Yamano, Ph.D.
Senior Researcher, Tokyo Metropolitan Institute of Medical Science

May 4th, 2020

RE: JCB Manuscript #201912144R

Dr. Koji Yamano
Tokyo Metropolitan Institute of Medical Science
Ubiquitin project
2-1-6 Kamikitazawa
Setagaya-ku, Tokyo 156-8506
Japan

Dear Dr. Yamano:

Thank you for submitting your revised manuscript entitled "Critical Roles of Ubiquitin Signals and the OPTN-ATG9A Axis in Mitochondria-Selective Autophagy" and for your patience as the peer review process was delayed by the current pandemic. We would be happy to publish your paper in JCB pending final revisions necessary to meet our formatting guidelines (see details below).

- Please consider the following alternative title suggestion to make the main advance accessible to as broad an audience as possible:

"Critical Role of Mitochondrial Ubiquitination and the OPTN-ATG9A Axis in Mitophagy"

- Provide the main and supplementary texts as separate, editable .doc or .docx files
- Provide main and supplementary figures as separate, editable files according to the instructions for authors on JCB's website *paying particular attention to the guidelines for preparing images and blots at sufficient resolution for screening and production*
- Format references for JCB
- Provide tables as excel files
- Add scale bars to figures 2F inset, 3B-E inset, 5A, B inset, 7E,G inset, 7H, S4A,B inset,
- Include tables in the Online Supplemental Materials paragraph

A. MANUSCRIPT ORGANIZATION AND FORMATTING:

Full guidelines are available on our Instructions for Authors page, <http://jcb.rupress.org/submission-guidelines#revised>. **Submission of a paper that does not conform to JCB guidelines will delay the acceptance of your manuscript.**

B. FINAL FILES:

-- High-resolution figure and video files: See our detailed guidelines for preparing your production-ready images, <http://jcb.rupress.org/fig-vid-guidelines>.

****It is JCB policy that if requested, original data images must be made available to the editors. Failure to provide original images upon request will result in unavoidable delays in publication. Please ensure that you have access to all original data images prior to final submission.****

****The license to publish form must be signed before your manuscript can be sent to production. A link to the electronic license to publish form will be sent to the corresponding author only. Please take a moment to check your funder requirements before choosing the appropriate license.****

Thank you for this interesting contribution, we look forward to publishing your paper in Journal of Cell Biology.

Sincerely,

Li Yu, Ph.D.
Monitoring Editor

Marie Anne O'Donnell, Ph.D.
Scientific Editor

Journal of Cell Biology

Reviewer #2 (Comments to the Authors (Required)):

The authors addressed all the questions from reviewers very well and I have no further comments.

Reviewer #3 (Comments to the Authors (Required)):

The authors have done an excellent job addressing the comments and have greatly strengthened the manuscript. Overall, this is a beautiful and very high quality study that makes an important

contribution to the field.